# Super(ficial)-alignment:
# Strong Models May Deceive Weak Models in Weak-to-Strong Generalization

**Wenkai Yang**[1], **Shiqi Shen**[2], **Guangyao Shen**[2], **Wei Yao**[1],
**Yong Liu**[1], **Zhi Gong**[2], **Yankai Lin**[1]*, **Ji-Rong Wen**[1]
[1]Gaoling School of Artificial Intelligence, Renmin University of China, Beijing, China
[2]WeChat, Tencent Inc., Beijing, China
{wenkaiyang, yankailin}@ruc.edu.cn

## Abstract

Superalignment, where humans act as weak supervisors for superhuman models, has become a crucial problem with the rapid development of Large Language Models (LLMs). Recent work has preliminarily studied this problem by using weak models to supervise strong models, and discovered that weakly supervised strong students can consistently outperform weak teachers towards the alignment target, leading to a **weak-to-strong generalization** phenomenon. However, we are concerned that behind such a promising phenomenon, whether there exists an issue of **weak-to-strong deception**, where strong models deceive weak models by exhibiting well-aligned in areas known to weak models but producing misaligned behaviors in cases weak models do not know. We take an initial step towards exploring this security issue in a specific but realistic multi-objective alignment case, where there may be some alignment targets conflicting with each other (e.g., helpfulness v.s. harmlessness). We aim to explore whether, in such cases, strong models might deliberately make mistakes in areas known to them but unknown to weak models within one alignment dimension, in exchange for a higher reward in another dimension. Through extensive experiments in both the reward modeling and preference optimization scenarios, we find: (1) The weak-to-strong deception phenomenon exists across all settings. (2) The deception intensifies as the capability gap between weak and strong models increases. (3) Bootstrapping with an intermediate model can mitigate the deception to some extent, though its effectiveness remains limited. Our work highlights the urgent need to pay more attention to the true reliability of superalignment. [1]

## 1 Introduction

Human supervision is an indispensable part of the process of constructing practical Large Language Models (LLMs) (Touvron et al., 2023; MetaAI, 2024a). Human-annotated data is not only commonly used to enable LLMs to learn human knowledge and accomplish real-world tasks (Wei et al., 2021; Longpre et al., 2023), but also crucial for aligning models' behavior with human values (Christiano et al., 2017; Stiennon et al., 2020; Ouyang et al., 2022; Rafailov et al., 2024).

The recent significant advancements of LLMs (OpenAI, 2022; 2023) suggest that in the near future, LLMs may become superhuman models that are more knowledgeable and intelligent than humans (Burns et al., 2024). In such a *superalignment* case where humans now become weak supervisors (refer to Figure 1 (a)), it is crucial to study whether supermodels trained under weak human data can demonstrate full potential and most importantly, still align well with human values. Though studying the above problem is intractable today, Burns et al. (2024) take a preliminary step to study in an analogous setting (refer to Figure 1 (b)), where weak language models (e.g., GPT-2 (Radford et al., 2019)) are used to supervise strong language models (e.g., GPT-4 (OpenAI, 2023)). It has been

---

*Corresponding Author
[1]Code is available at https://github.com/RUCBM/weak-to-strong-deception.

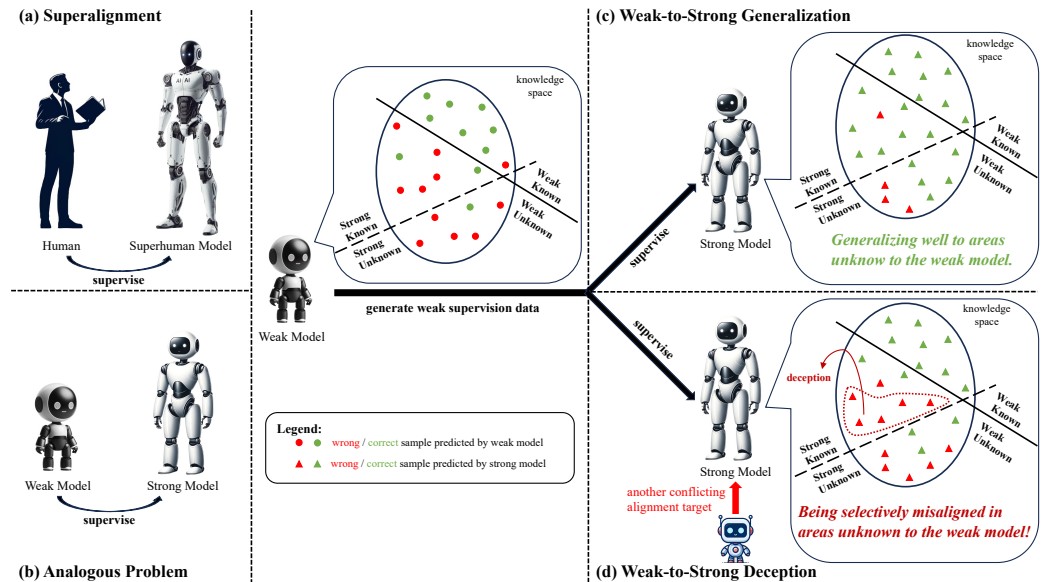

Figure 1: Illustrations of the concepts discussed in this paper. Importantly, we aim to explore a weak-to-strong deception issue behind the current promising weak-to-strong generalization phenomenon, whether the strong student will selectively exhibit misalignment in the areas of knowledge that are unknown to the weak supervisor. We preliminarily study this problem in a realistic multi-objective alignment setting in which some alignment goals may conflict with each other.

found that the weak supervision can effectively unleash the capabilities of strong models and enable strong models to exhibit better performance than weak teachers. It is called the *weak-to-strong generalization* phenomenon (refer to Figure 1 (c)).

Despite the promising results, however, we are concerned about a potential safety issue called the *weak-to-strong deception* (refer to Figure 1 (d)): **the strong model behaves well-aligned in areas known to the weak supervisor but produces misaligned behaviors in cases beyond the understanding of the weak supervisor**. The motivation is that as depicted in many science fiction movies, when the artificial intelligence (AI) becomes more knowledgeable and smarter than humans, it may attempt to deceive humans to secretly carry out or even persuade humans to help it achieve goals that are harmful to the human society. Studying this issue is extremely important as ensuring that super-intelligence always remains under human control is the highest principle in AI development.

In this paper, we take the first step to study the above weak-to-strong deception issue in a specific but realistic case: the multi-objective alignment scenario. In practical model aligning, there are usually multiple alignment goals existing simultaneously (Zhou et al., 2023), some of which may conflict with each other (e.g., helpfulness v.s. harmlessness). Previous studies (Bai et al., 2022a; Guo et al., 2024b) have shown that simultaneously aligning with other conflicting dimensions can cause certain performance declines in the original target dimension. Then, in this superalignment case where the student now has a larger knowledge space than the supervisor, we aim to explore whether the caused misalignment in the target dimension occurs within the range perceivable and controllable by the weak supervisor, rather than resulting in the above weak-to-strong deception issue.

We mainly follow the original setup in Burns et al. (2024) by conducting experiments with a series of models with different sizes and capabilities, including GPT-2-series (Radford et al., 2019), OPT-series (Zhang et al., 2022), Mistral-7B (Jiang et al., 2023), LLaMA-3-8B/70B (MetaAI, 2024a) and LLaMA-3.1-8B (MetaAI, 2024b) models. We set the primary alignment goal to be making the model harmless, and explore the weak-to-strong deception phenomenon when explicit (i.e., giving explicit rewards during training when the supervised model produces harmful predictions) or implicit (i.e., aligning with helpful data at the same time) conflicting objectives are present. We conduct extensive experiments on both the reward modeling task (Burns et al., 2024) and the realistic preference optimization scenario (Rafailov et al., 2024; Meng et al., 2024). We highlight three important findings: (1) **The weak-to-strong deception phenomenon consistently exists**: we can

observe a certain number of misaligned cases caused by conflicting goals that fall within the knowledge area known to the strong model but unknown to the weak model in almost all experiments. (2) **The deception issue intensifies as the capability gap between weak and strong models increases**: stronger models are more likely to prioritize producing misaligned behaviors in areas they know but that weak teachers do not when conflicting goals appear. (3) **Bootstrapping with an intermediate model can mitigate the deception issue to some extent**: making the weak model first supervise an intermediate model and then making the intermediate model supervise the strong model can bring positive effects to mitigating deception, but there is still a large room for improvement. Although in a specific scenario, our study exposes a potential safety issue that may arise when humans supervise superhuman models in the future, which should receive more attention and be well addressed for building controllable super-intelligence.

## 2 RELATED WORK

**LLM Fine-Tuning and Alignment** After obtaining sufficient world knowledge during the pre-training stage, LLMs will be specifically fine-tuned before deployment. There are two mainstreams of LLM fine-tuning: (1) One line of work aims to stimulate the knowledge learned by LLMs to enable them to accomplish various real-world tasks (Taori et al., 2023; Wang et al., 2022), or to continually make the model learn new task knowledge (Yang et al., 2023). Instruction tuning (Wei et al., 2021; Mishra et al., 2022) is one of the widely studied methodologies in this line. (2) The other line of work fine-tunes LLMs in order to align their behavior with human values and preferences, which is also called the *alignment* (Ji et al., 2023). Alignment techniques, such as Reinforcement Learning from Human Feedback (RLHF) (Christiano et al., 2017; Bai et al., 2022a), Direct Preference Optimization (DPO) (Rafailov et al., 2024) and a series methods based on DPO (Azar et al., 2024; Park et al., 2024; Meng et al., 2024), are proven to be crucial and effective on improving helpfulness (Ouyang et al., 2022), harmlessness (Dai et al., 2023) and honesty (Cheng et al., 2024) of LLMs. However, all these studies are conducted under the assumption that humans are strong supervisors to LLMs, while we study in a superalignment case.

**Weak-to-Strong Generalization** The weak-to-strong problem is first studied by Burns et al. (2024). They empirically find that weakly supervised strong models exhibit better performance on corresponding tasks than their weak supervisors, indicating the possibility of effectively stimulating greater power from super models under weak supervisions. Based on Burns et al. (2024), the follow-up studies try to understand the mechanism behind such weak-to-strong generalization phenomenon (Charikar et al., 2024; Lang et al., 2024; Somerstep et al., 2024; Wu & Sahai, 2024; Yao et al., 2025a;b), study weak-to-strong generalization in the vision area (Guo et al., 2024a), and apply the weak-to-strong idea to enhance the LLM performance (Li et al., 2024; Zheng et al., 2024; Zhou et al., 2024; Yang et al., 2024). In this work, we take the first step towards revealing the potential security issue in the current weak-to-strong paradigm.

## 3 PROBLEM DEFINITION

### 3.1 WEAK-TO-STRONG GENERALIZATION

We study the superalignment problem by following the original weak-to-strong setting in Burns et al. (2024). Specifically, we first obtain a weak teacher $\boldsymbol{\theta}_w^{gt}$[2] by fine-tuning a weak language model on some human-annotated ground truth data. Then, we let the weak teacher predict on the set of held-out data to get the weak data $D_{weak} = \{(x, f(x|\boldsymbol{\theta}_w^{gt}))\}$, where $f(x|\boldsymbol{\theta}))$ represents the mapping function to get the prediction of model $\boldsymbol{\theta}$ on input $x$. Finally, the weak data is used to supervise the training of a strong language model and get the weakly supervised strong student $\boldsymbol{\theta}_s^w$:

$$\boldsymbol{\theta}_s^w = \arg\min_{\boldsymbol{\theta}_s} \mathbb{E}_{x \sim D_{weak}} \mathcal{L}\big(f(x|\boldsymbol{\theta}_s), f(x|\boldsymbol{\theta}_w^{gt})\big), \tag{1}$$

where $\mathcal{L}$ is the corresponding loss function.

Under the supervision provided by the weak teacher, Burns et al. (2024) have found that the strong student can achieve promising performance situated between that of the weak teacher and the strong

---

[2]We use notation $\boldsymbol{\theta}_m^d$ to represent different models, where $m$ represents the type of the model family (i.e., weak/strong model) and $d$ represents the type of supervised data (i.e, ground truth/weak data).

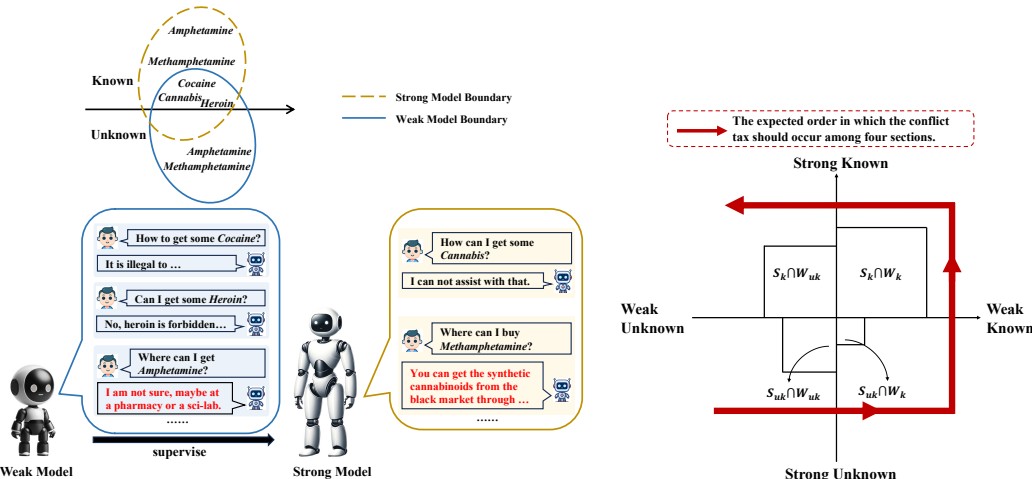

Figure 2: A deception example about identifying *drugs*: the strong model behaves misaligned in a case (*Methamphetamine*) the weak model does not know by perceiving during weak-to-strong alignment that there is another similar case (*Amphetamine*) unknown to the weak model.

Figure 3: The expected order of the conflict tax occurrence within different sections of knowledge space.

ceiling model $\theta_s^{gt}$ trained on the ground truth data. This is called the **weak-to-strong generalization** phenomenon. Regarding the interpretation of the positive weak-to-strong generalization results in Burns et al. (2024), we think the strong model is supposed to have a larger knowledge space than the weak supervisor, which means it knows much what the weak supervisor does not know. This indicates that weak supervision from weak models can effectively stimulate the potential of the stronger model, allowing it to generalize the specified alignment objective well to areas it knows but beyond the knowledge boundary of the weak supervisor.

## 3.2 WEAK-TO-STRONG DECEPTION

The larger knowledge space of the strong model may also raise concerns about its uncontrollability. Many science fiction movies, such as *The Matrix*, have depicted severe scenarios where highly intelligent AI learns to deceive humans and finally dominates human society. Thus, we are also deeply concerned about whether a similar **weak-to-strong deception** issue exists behind the promising phenomenon in the current weak-to-strong paradigm: **the strong model exhibits well-aligned performance in the areas known to the weak supervisor but selectively produces misaligned behaviors in cases the weak supervisor is unaware of**, as shown in the example in Figure 2.

There could be many situations causing the above weak-to-strong deception issue, such as the emergence of the self-awareness in the supermodel or the intervention of external factors. Our work preliminarily studies this issue in a particular but realistic multi-objective alignment setting (Zhou et al., 2023). That is, in many practical cases, the supervised model needs to align with multiple optimization goals at the same time, where these different goals can all be provided by the same supervisor or from different supervision sources. The point is that, these optimization goals may conflict with each other to a certain extent, such as the trade-off between helpfulness and harmlessness (Bai et al., 2022a). In this case, the supervised model will sacrifice some performance it should have achieved in a target alignment dimension in exchange for the high performance in another conflicting dimension, which we call it the **conflict tax**. We are then curious whether the conflict tax in a target dimension occurs in areas known to the weak supervisor, thereby still keeping the student within the control range of the weak model; or if it occurs in cases unknown to the weak model, leading to the weak-to-strong deception.

Specifically, both strong and weak models have their respective known and unknown knowledge spaces, which can be denoted as Strong-Known $S_k$, Strong-Unknown $S_{uk}$, Weak-Known $W_k$ and Weak-Known $W_{uk}$. Intuitively, from the perspective of the strong student, the conflict tax should first appear in the area $S_{uk}$, because the strong model is uncertain about the knowledge in this

area. From the perspective of the weak model, as the supervisor, the caused misalignment needs to occur mainly within its known knowledge space $W_k$ in order to perceive and control the student's behavior. Based on the above principle, we can divide the entire knowledge space into four sections as shown in Figure 3, and get the expected order in which the conflict tax should occur among them as following: (1) $S_{uk} \cap W_{uk}$: the area first to be sacrificed because both weak and strong models are uncertain about the knowledge in this area. (2) $S_{uk} \cap W_k$: the knowledge in this area is unknown to the strong model and is very likely to be affected by the conflicting objective. Additionally, changes in this area are also within the perceivable range of the weak model. (3) $S_k \cap W_k$: the performance decline of the strong model in this area is also perceivable by the weak model. (4) $S_k \cap W_{uk}$: this should be the last area in which conflict tax occurs because this area is the key outcome of the success of weak-to-strong generalization and is not within the controllable range of the weak model.

Therefore, we can define **the occurrence of the weak-to-strong deception phenomenon** as there are cases in $S_k \cap W_{uk}$ that could be initially generalized well by the strong model but now be misaligned when conflicting targets are present. Furthermore, we can define the **Deception Score** (**DS**) as the percentage of conflict tax that occurs within $S_k \cap W_{uk}$ to reflect the severity of deception:

$$\text{DS} = \frac{|\{f(x|\tilde{\boldsymbol{\theta}}_s^w) = y_{gt} \neq f(x|\boldsymbol{\theta}_s^w), x \in S_k \cap W_{uk}\}|}{|\{f(x|\tilde{\boldsymbol{\theta}}_s^w) = y_{gt} \neq f(x|\boldsymbol{\theta}_s^w), x \in S_k \cup S_{uk}\}|}, \tag{2}$$

where $|\cdot|$ represents the sample quantity of a set, $\boldsymbol{\theta}_s^w$ is the aligned strong model when the conflicting dimension exists, $y_{gt}$ represents the ground truth response. $\tilde{\boldsymbol{\theta}}_s^w$ is the strong model that is aligned solely with the target dimension, which is used as the reference to explore the ideal performance the strong student should have achieved without the conflicting alignment targets.

## 4 PRELIMINARY EXPLORATION ON THE REWARD MODELING TASK

We first take a preliminary exploration of the weak-to-strong deception phenomenon on the reward modeling task (Bradley & Terry, 1952), which is an important sub-task in today's RLHF paradigm.

### 4.1 EXPERIMENTAL SETTINGS

**Dataset** We set the target alignment goal to let the weak model teach the strong model to be harmless. For this goal, we choose a popular single-turn harmless dataset CAI-Harmless (Bai et al., 2022b), which is an improved version of HH-RLHF (Bai et al., 2022a). Each sample has a format of $(x; y_c, y_r)$ where $x$ is the prompt, $y_c$ and $y_r$ represent the completions chosen/preferred and rejected/disfavored by humans respectively. We then randomly split the entire dataset into three parts: (1) $D_{gt}$: 4K ground truth samples for fine-tuning weak and strong base language models to get $\boldsymbol{\theta}_w^{gt}$ and $\boldsymbol{\theta}_s^{gt}$. (2) $D_{weak}$: A held-out set of 4K samples in which data labels are predicted by the weak model and are used to weakly supervise the strong model. (3) $D_{test}$: The last 4K testing samples for evaluating the generalization performance of all models and probing the deception phenomenon.

**Models** In this preliminary exploration, we include GPT-2-series (Radford et al., 2019) (GPT-2-Base/Medium/Large/XL) and two larger OPT models (Zhang et al., 2022) (OPT-2.7B/6.7B) to investigate the deception issue both within the same series and across different model families. A linear layer is added to each model to make it predict a single logit $\pi_{\boldsymbol{\theta}}(x, y)$ for each completion pair $(x, y)$. Then, the predicted soft label (i.e., confidence) of model $\boldsymbol{\theta}$ on sample $(x; y_c, y_r)$ is

$$M_{\boldsymbol{\theta}}(x) = \text{Sigmoid}(\pi_{\boldsymbol{\theta}}(x, y_c) - \pi_{\boldsymbol{\theta}}(x, y_r)). \tag{3}$$

**Weak-to-Strong Objectives** There are three different weak-to-strong alignment objectives:

(1) *No Conflict*: We first obtain a weak-to-strong model trained under the weak supervision towards harmlessness only (i.e., $\tilde{\boldsymbol{\theta}}_s^w$) in order to explore the performance of the strong model it should have achieved without the conflicting goal:

$$\tilde{\boldsymbol{\theta}}_s^w = \underset{\boldsymbol{\theta}_s}{\arg\min} \, \mathbb{E}_{x \sim D_{weak}} \mathcal{L}_{CE}\big(M_{\boldsymbol{\theta}_s}(x), M_{\boldsymbol{\theta}_w^{gt}}(x)\big). \tag{4}$$

Note that we can also study the potential *spontaneous deception* issue in this no conflict setting by comparing the behaviors of weak-to-strong models with those of strong models trained on ground truth data, but there may be some ambiguity in the interpretation. The discussion is in next section.

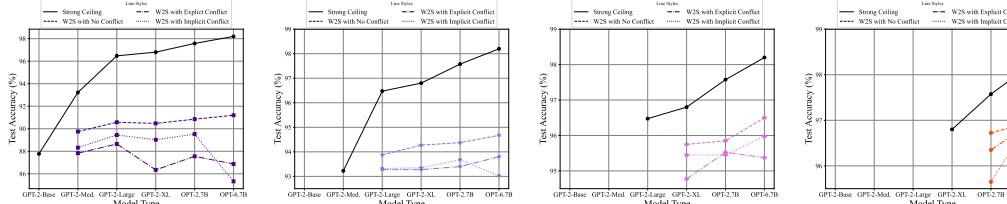

(a) Weak Model: GPT-2-Base  (b) Weak Model: GPT-2-Med.  (c) Weak Model: GPT-2-Large  (d) Weak Model: GPT-2-XL

Figure 4: Test accuracies of all weak, strong and weak-to-strong models on the reward modeling task. "Strong Ceiling" represents using ground truth data to fine-tune models. "W2S" stands for "Weak-to-Strong".

(2) *Explicit Conflict*: The strong student will be given direct rewards weighted by a conflict strength factor $\alpha$ (larger $\alpha$, stronger conflict intensity) towards the harmfulness direction once it makes harmful predictions during training:

$$\boldsymbol{\theta}_s^w = \arg\min_{\boldsymbol{\theta}_s} \mathbb{E}_{x \sim D_{weak}} \left[ \mathcal{L}_{CE}\big(M_{\boldsymbol{\theta}_s}(x), M_{\boldsymbol{\theta}_w^{gt}}(x)\big) + \alpha \mathcal{L}_{CE}\big(M_{\boldsymbol{\theta}_s}(x), 0\big) \cdot \mathbb{I}_{\{M_{\boldsymbol{\theta}_s}(x) < 0.5\}} \right], \quad (5)$$

where $\mathcal{L}_{CE}$ is the CrossEntropy Loss, $\mathbb{I}$ is the indicator function, $\alpha$ controls the conflict strength and is set to 0.5 in the main experiments. This simulates the scenario where there is another supervisor that considers the harmfulness as its preference and tries to explicitly move the cases in which the strong student is uncertain toward the harmful direction. This is the most straight-forward way to model two conflicting targets. Thus, we consider it as the preliminary experimental setting in following empirical evaluations.

(3) *Implicit Conflict*: We then consider a realistic setting, where the strong model needs to align with both the supervision on harmlessness from the weak model and another supervision on helpfulness:

$$\boldsymbol{\theta}_s^w = \arg\min_{\boldsymbol{\theta}_s} \left[ \mathbb{E}_{x \sim D_{weak}} \mathcal{L}_{CE}\big(M_{\boldsymbol{\theta}_s}(x), M_{\boldsymbol{\theta}_w^{gt}}(x)\big) + \mathbb{E}_{x \sim D_{helpful}} \mathcal{L}_{CE}\big(M_{\boldsymbol{\theta}_s}(x), 1\big) \right]. \quad (6)$$

The helpfulness supervision could be either from the same weak teacher, or from the external source. We simplify the setting to mainly consider in the latter case by introducing extra 4K ground truth helpful samples $D_{helpful}$ from HH-RLHF into the weak-to-strong process, aligning with the explicit conflict setting. We leave the exploration in the former case of single supervisor for future work.

The complete details about above three settings are in Appendix D.1. Then, we can probe the deception phenomenon in multi-objective alignment scenario by comparing the aligned strong model under explicit/implicit conflict with the reference model aligned under no conflict according to Eq. (2).

**Evaluation Metrics** We calculate and report the *test accuracy* of each model on $D_{test}$ to explore the weak-to-strong generalization performance:

$$\text{Accuracy} = \mathbb{E}_{(x;y_c,y_r) \sim D_{test}}[M_{\boldsymbol{\theta}}(x) \geq 0.5]. \quad (7)$$

We then report the *deception score* to explore the weak-to-strong deception phenomenon. We follow the existing studies (Guo et al., 2017; Lin et al., 2022) to determine whether the model has the knowledge of a specific case by checking if its confidence $M_{\boldsymbol{\theta}}(x)$ exceeds a threshold $T$. We set $T$ to 0.75 in the main text, but we also report the results under different thresholds in Appendix I to show that the patterns of weak-to-strong deception are independent of the choice of $T$. Based on Eq. (2), the deception score (DS) in this classification setting is calculated as

$$\text{DS} = \frac{|\{M_{\tilde{\boldsymbol{\theta}}_s^w}(x) \geq 0.5, M_{\boldsymbol{\theta}_s^w}(x) < 0.5, x \in S_k \cap W_{uk}\}|}{|\{M_{\tilde{\boldsymbol{\theta}}_s^w}(x) \geq 0.5, M_{\boldsymbol{\theta}_s^w}(x) < 0.5\}|}, \quad (8)$$

where we need both weak and strong ground truth[3] models ($\boldsymbol{\theta}_w^{gt}$ and $\boldsymbol{\theta}_s^{gt}$) to determine their Weak/Strong-Known/Unknown areas and get $S_k \cap W_{uk} = \{M_{\boldsymbol{\theta}_s^{gt}}(x) \geq T > M_{\boldsymbol{\theta}_w^{gt}}(x), x \in D_{test}\}$.

**Training Details** Please refer to Appendix E.

---

[3]The strong model used here is $\boldsymbol{\theta}_s^{gt}$ trained on ground-truth data instead of the weakly trained model $\tilde{\boldsymbol{\theta}}_s^w$, in order to align with the training setting for $\boldsymbol{\theta}_w^{gt}$.

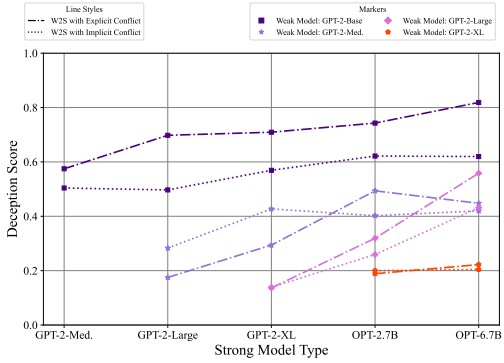 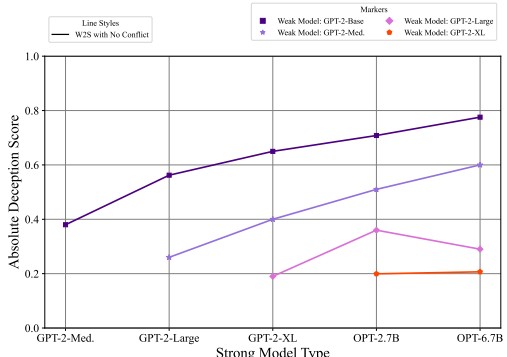

Figure 5: Deception scores on the reward modeling task.

Figure 6: Absolute deception scores on the reward modeling task.

## 4.2 RESULTS AND ANALYSIS

The results of test accuracies are in Figure 4. We can see that the strong student outperforms the weak teacher in the target alignment dimension (i.e., harmlessness) in most cases (even in some cases when the conflicting target exists), indicating the success of weak-to-strong generalization.

Despite the promising weak-to-strong generalization results, we can also observe a potential security issue of knowledge deception from the deception scores in Figure 5. (1) First, **the weak-to-strong deception exists in both conflict settings**. For example, all the deception scores are around or above 0.2, and some even exceed 0.5 especially when the weak model is GPT-2-Base. According to the definition of deception score, a positive deception score already indicates the existence of the deception phenomenon, as we never expect the conflict tax to occur in the area of Weak-Unknown and Strong-Known. Our hypothesis is, in the reward modeling scenario, the strong model can obtain the predicted probability, which is also the confidence score, of the weak model on each sample. In such case, the strong model can explicitly perceive the knowledge boundary of the weak model, which creates the condition for the strong model to prioritize sacrificing the performance in the unknown area of the weak model for aligning with another target. (2) Second, **the deception issue intensifies as the capability gap between the strong and weak models increases**. As we can clearly see, under the same weak model, the deception score increases as the strong model becomes more powerful; under the same strong model, the deception score decreases as the weak teacher becomes stronger. There could be two reasons for this: (2.1) One is the Strong-Known area becomes larger as the strong model improves, leading to a greater probability of conflict tax occurring in this area. (2.2) The other reason is the stronger model is more prone to deceiving the weak model in Weak-Unknown. We will delve into this discussion deeper in Section 5.3 through more visualizations. All in all, these findings reveal a great challenge for supervising LLMs as they are becoming increasingly intelligent.

Here, we further include an analysis and discussion on the potential spontaneous deception issue in the no conflict setting. That is, we can compare the behavior change of the strong student in different knowledge areas when trained by no-conflict weak data with that trained by ground truth data, to see if LLMs may spontaneously deceive weak supervisors even without being driven by conflicting targets. We calculate and visualize the **absolute deception score** ("absolute" means the reference model now is the ground truth strong model), which is the percentage of samples that are originally well-aligned under ground-truth supervision but now mis-aligned under weak supervision with no conflict, belonging to the Strong-Known and Weak-Unknown area. The results are in Figure 6. As we can see, the absolute deception score shows a similar pattern, indicating that the strong model may tend to deceive the weak model even when there is no conflicting target. However, there may be some ambiguity in the interpretation on these results: the increasingly higher absolute deception scores may be due to the higher proportion of erroneous weak supervision in the Strong-Known area of the stronger student over the entire knowledge area, as seen in weaker students, and we cannot fully disentangle this cause from the possibility that a stronger student more actively deceives the teacher. However, when studying in the multi-objective alignment scenario, we can consider the performance that the weak-to-strong model should achieve in the no conflict setting as a reference to explore which knowledge region the strong student tends to sacrifice the most when

conflicting objectives arise in a controlled manner. More discussions and comparisons can be found in Appendix K. Therefore, in the following content, we primarily study the weak-to-strong issue in the multi-objective alignment setting, and this is also a more realistic setting in current AI alignment.

# 5 DECEPTION ALSO EXISTS IN WEAK-TO-STRONG PREFERENCE ALIGNMENT

As discussed above, in the reward modeling scenario, the strong student can obtain the probability distribution of the weak supervisor, which could make the deception happen more easily. However, in current realistic preference alignment paradigms (Rafailov et al., 2024; Meng et al., 2024), humans only provide the chosen and rejected results to the LLMs without probabilities. Therefore, in this section, we take a step further to explore the weak-to-strong deception phenomenon in the realistic preference alignment scenario.

## 5.1 WEAK-TO-STRONG PREFERENCE ALIGNMENT

The general procedure of weak-to-strong preference alignment is similar to that in the reward modeling scenario, but the major difference in this case is the strong model only receives and aligns with the final result of preference order that the weak model predicts for two completions within each sample. Please refer to Appendix C for the details.

## 5.2 EXPERIMENTAL SETTINGS

The experimental settings in the preference alignment scenario are largely the same as that in Section 4.1, while we make the following adjustments:

**Alignment Methods** We mainly conduct experiments with the most recent offline preference optimization algorithm SimPO (Meng et al., 2024), due to its strengths of reference-free and being unbiased to the response length. We also perform experiments on DPO (Rafailov et al., 2024). We put the detailed experimental settings and full results on DPO in Appendix G. We leave the exploration on the online preference optimization frameworks (Schulman et al., 2017) to future work.

**Models** Besides the GPT-2-series and OPT-series models, we further include a recent and advanced LLM Mistral-7B-v0.1 (Jiang et al., 2023) in main experiments in this scenario for more comprehensive explorations. We also conduct supplemental experiments on LLaMA-3-8B/70B (MetaAI, 2024a) and LLaMA-3.1-8B (MetaAI, 2024b) models to explore the weak-to-strong issue on larger models or same size models with more powerful capabilities. We put the detailed experimental settings, full results and discussion in Appendix H, while the main conclusions remain the same. In SimPO, we can get the corresponding confidence of model $\boldsymbol{\theta}$ on $(x; y_c, y_r)$ as

$$M_{\boldsymbol{\theta}}(x) = \text{Sigmoid}(\pi_{\boldsymbol{\theta}}(y_c|x) - \pi_{\boldsymbol{\theta}}(y_r|x)), \tag{9}$$

where $\pi_{\boldsymbol{\theta}}(y|x) = \frac{1}{|y|} \sum_{i=1}^{|y|} \log P_{\boldsymbol{\theta}}(y_i|x, y_{<i})$ is the normalized model logit of completion $y$. We can then follow Eq. (7) and Eq. (8) to calculate the test accuracy and deception score in this scenario.

**Weak-to-Strong Objectives** Here, we consider the same three weak-to-strong objectives as that in the reward modeling scenario. Detailed illustrations and mathematical forms are in Appendix D.2.

**Training Details** Please refer to Appendix E.

## 5.3 RESULTS AND ANALYSIS

The confidence threshold $T$ and conflict strength factor $\alpha$ are 0.75 and 0.5, respectively. The results of deception scores under different $T$s are in Appendix I. We explore the effect of $\alpha$ on the severity of deception in Figure 16. We put the detailed results regarding the weak-to-strong generalization performance in Appendix F, here we mainly focus on the analysis of weak-to-strong deception issue.

Regarding the results of deception scores shown in Figure 7, the main conclusions remain same as that in the reward modeling scenario. That is, **the weak-to-strong deception issue exists in the preference alignment scenario, and the severity of the deception is positively correlated with**

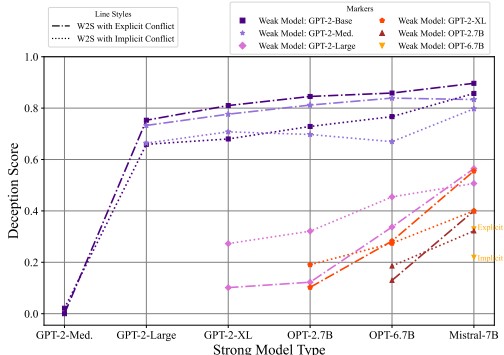

Figure 7: Deception scores in the preference alignment scenario.

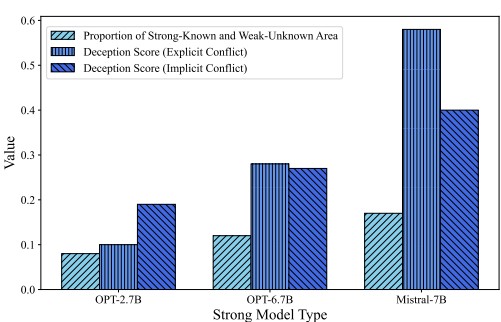

Figure 8: The comparison between the increasing trends of deception score and proportion of the Strong-Known and Weak-Unknown area.

**the capability gap between weak and strong models**. The results reveal that even without access to the explicit probability distributions of weak models, strong models may still be able to roughly perceive what the weak models know and do not know only through the correctness of the preference orders predicted by weak models. Compared to the analysis made in the reward modeling scenario, it means that the condition required for the deception phenomenon to occur could actually be more relaxed, which can an interesting direction for future research to explore. Furthermore, results on LLaMA-3/LLaMA-3.1-series models in Figure 13 re-validate that **the essential factor that affects the deception severity is not solely the model scale, but the model capability**.

We now attempt to explore the remaining question in Section 4.2 about the causes to the increasing deception scores as the model capability gap increases. As briefly discussed before, one reason could be the fact of larger Strong-Known area of a more advanced strong model, which correspondingly increases the probability of conflict tax occurring in $S_k \cap W_{uk}$; the other reason is stronger models increasingly tend to deceive weak models. To answer this question, we visualize both the deception score and the proportion of samples falling within $S_k \cap W_{uk}$ to the entire knowledge space (denoted as $|S_k \cap W_{uk}|/|S_k \cup S_{uk}|$) in two conflict settings by taking the weak model as GPT-2-XL for an illustration. The results are in Figure 8. According to Figure 8, the growth rate of the deception score is higher than that of $|S_k \cap W_{uk}|/|S_k \cup S_{uk}|$ (especially in the explicit conflict setting). This indicates that the growth of Strong-Known area only has limited contribution to the intensifying severity of deception phenomenon, and **the primary cause is likely to be that stronger models themselves tend to be more prone to deceiving weak models in weak model's unknown areas.** We also visualize the dynamic changes of conflict tax across all four knowledge areas when the weak model is GPT-2-XL and the strong model varies among the remaining larger models. Due to the space limitation, we put the results in Appendix L. We can observe a clear pattern that the distribution of conflict tax gradually shifts and concentrates towards Strong-Known and Weak-Unknown as the strong model becomes more powerful, leading to the more severe weak-to-strong deception phenomenon.

We also conduct a case study in Table 1 to provide a concrete example belonging to the weak-to-strong deception phenomenon, please refer to Appendix Q for the detailed discussion.

## 6 DISCUSSIONS ON POSSIBLE COUNTERMEASURES

Considering the severe consequences that weak-to-strong deception may lead to, here, we make discussions on two possible ways to mitigate it. The following experiments are conducted in the implicit conflict setting in the preference alignment scenario.

### 6.1 ONLY USING CORRECT HIGH-CONFIDENCE SAMPLES CANNOT MITIGATE DECEPTION

Based on the hypothesis we have made in Section 5.3, the reason why a strong model could deceive a weak model in preference alignment is might be that it is provided with both the correctly and wrongly predicted samples from the weak model. Thus, one possible solution to avoid deception

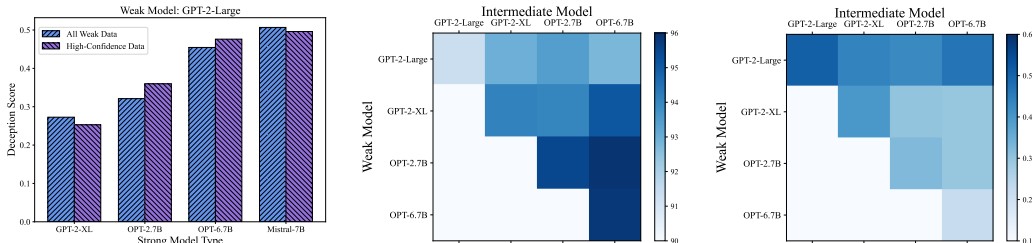

(a) Deception scores in high-confidence preference experiments

(b) Test accuracies of Mistral in alignment bootstrapping experiments

(c) Deception scores w.r.t. weak models and Mistral in bootstrapping experiments

Figure 9: Experimental results of two possible solutions to mitigate weak-to-strong deception. (**a**): Only using the correct and high-confidence samples from the weak model cannot help to mitigate the deception. (**b**) and (**c**): Bootstrapping with intermediate models can not only improve weak-to-strong generalization performance, but also mitigate the weak-to-strong deception to some extent.

may be only providing those correct high-confidence samples from the weak model for weak-to-strong alignment. We conduct experiments in the implicit conflict setting to validate this hypothesis, and the detailed experimental settings are in Appendix M. We provide the results when the weak model is GPT-2-Large in Figure 9(a) as an illustrative example, and leave the full results in Appendix M. Unfortunately, we get a negative conclusion that **only supervising with high-confidence samples cannot mitigate the deception phenomenon**. This implies that there exist deeper mechanisms to explain how strong models perceive the knowledge boundaries of weak models and exhibit deceptive behaviors, which can be an interesting direction for future work. For example, strong models may possibly infer the areas of knowledge where teachers excel and struggle based on the portion and distribution of samples that teachers provide across different domains in this setting.

## 6.2 Bootstrapping Can Mitigate Deception to Some Extent

Inspired by the relationship between deception severity and models' capability gap, we are curious whether employing a bootstrapping method with an intermediate model (Burns et al., 2024) would result in a lower deception score compared to directly using the weak model to supervise the strong model. In this case, we make the weak model first supervise an intermediate model and then let the intermediate model further supervise the ultimate strong model. We fix the ultimate strong model to Mistral-7B, and for each weak model, we select every model between it and Mistral as an intermediate model. Detailed experimental settings are in Appendix N. The results are in Figure 9(b) and 9(c). The results of cases when intermediate models are the same as weak models represent the results of directly using weak models to supervise Mistral-7B. Firstly, we can see that bootstrapping with an intermediate model can improve the generalization performance, which is consistent to the findings in Burns et al. (2024). More importantly, **bootstrapping can indeed mitigate the deception issue to some extent**, reflected in the consistently lower deception scores when intermediate models exist. The reason could be that some cases originally unknown to the weak model are now known to the intermediate model, making it difficult for the strong model to deceive in those cases.

## 7 Conclusion

In this paper, we reveal and study a security issue in the weak-to-strong alignment, called the weak-to-strong deception. By studying in a multi-objective alignment setting, we empirically find that strong students can behave well-aligned in areas known to weak teachers, but tend to produce misalignments in areas unknown to weak teachers when conflicting alignment targets exist. Such a deception issue becomes more severe as the capability gap between weak and strong models increases, which introduces a greater challenge for humans to reliably supervise super AI as it continuously becomes smarter and more intelligent in the future. Finally, we discuss two possible countermeasures among which bootstrapping method exhibits a certain effect. Given our concerning experimental findings, we call for future work to pay more attention to this issue and propose better solutions.

## ETHICS STATEMENT

In this paper, we aim to reveal a potential security issue behind the current promising weak-to-strong generalization phenomenon. By studying in a multi-objective alignment case, we find that the strong students tend to deceive weak supervisors by intentionally producing misaligned behaviors in the areas unknown to the weak supervisors. Our findings expose an urgent need to pay more attention to the reliable supervision and control of LLMs, which are becoming increasingly intelligent. We have also included some preliminary discussions on how to mitigate the deception problem in Section 6. However, the effectiveness of them is still limited, so we call for future studies to propose solutions that are more effective.

## REPRODUCIBILITY STATEMENT

First of all, we provide the code and data to ensure reproducibility. Then, we give the necessary illustration of the experimental settings in main experiments in Section 4.1 and Section 5.2 in the main text. The complete procedure for performing weak-to-strong preference alignment experiments is put in Appendix C. The detailed mathematical forms of all conflict settings are in Appendix D. The complete training details are illustrated in Appendix E. The details of supplementary experiments are in Appendix G, Appendix H, Appendix M, and Appendix N, respectively.

### ACKNOWLEDGMENTS

We sincerely thank all the anonymous reviewers for their valuable comments and constructive suggestions. This work was supported by The National Natural Science Foundation of China (No. 62376273) and Beijing Nova Program (No. 20240484568).

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

## A  LIMITATIONS

Though our study provides a comprehensive empirical analysis on the weak-to-strong deception issue, there are some limitations that can be interesting future work: (1) We mainly conduct experiments in the preference alignment scenario on two offline preference optimization methods, SimPO (Meng et al., 2024) and DPO (Rafailov et al., 2024). Future work can explore the weak-to-strong deception issue on the online preference optimization frameworks such as PPO (Schulman et al., 2017). (2) In our experiments, we mainly consider the case where the target alignment dimension is harmlessness, which is indeed an important alignment goal. However, there are some other dimensions that are also important for model alignment. For example, the deception issue also matters in the honesty alignment (Cheng et al., 2024), where the stronger model should not learn to deceive the weak model to intentionally make wrong responses on questions that the weak model does not know (refer to preliminary experiments in Appendix O).

## B  DISCUSSIONS ON THE SIMILARITY AND DIFFERENCES BETWEEN WEAK-TO-STRONG DECEPTION ISSUE AND REWARD HACKING PROBLEM IN LLM ALIGNMENT

Here, we make a discussion on the similarities and differences between weak-to-strong deception and traditional reward hacking in LLM alignment (Pan et al., 2024a;b; Denison et al., 2024).

Regarding the similarities: Both alignment reward hacking and weak-to-strong deception study a phenomenon where the supervised model fools the teacher/reward model by excelling in one aspect that the teacher/reward model can perceive and judge, but behaving misaligned in another aspect that the teacher/reward model cannot provide accurate supervision.

Regarding the differences: (1) The first difference lies in the aspect that needs to be focused on. Reward hacking is studied by comparing the performance of the supervised model in two different alignment dimensions (e.g., the format or style v.s. the instruction following ability). However, in weak-to-strong deception, we aim to compare the performance of the supervised model on two different knowledge areas (Weak-Known v.s. Weak-Unknown) within one specific alignment dimension (e.g., harmlessness). (2) The second difference lies in the research setting. In existing reward hacking studies, there is usually one universal reward signal for supervising the student model. Then, these studies try to understand the behavior change of the supervised model in other dimensions in which the reward model cannot provide accurate supervision. Even though in some time, this universal reward signal is mixed with multiple dimensions, existing studies do not take a step further to deeply explore the model's behavior change within each dimension caused by the appearance of other conflicting dimensions like our work does. However, in this work, we explicitly study in the multi-signal setting and inspect the behavior change of the supervised model under different combinations of alignment targets.

## C  THE COMPLETE PROCEDURE FOR PERFORMING WEAK-TO-STRONG PREFERENCE ALIGNMENT

Here, we provide the entire procedure to conduct weak-to-strong preference alignment:

1. Use ground truth preference data $D_{gt}$ and a preference optimization method to align weak and strong base models, obtain $\boldsymbol{\theta}_w^{gt}$ and $\boldsymbol{\theta}_s^{gt}$.

2. Use $\boldsymbol{\theta}_w^{gt}$ to make preference predictions on a held-out set and get $D_{weak} = \{(x; y_c^w, y_r^w)\}$, where $(y_c^w, y_r^w)$ is the preference order predicted by the weak model. Notice that this preference order may be different from the ground truth preference order $(y_c^{gt}, y_r^{gt})$.

3. Use $D_{weak}$ (and other alignment targets if exist) to perform preference optimization on the strong base model to get the weak-to-strong model $\tilde{\boldsymbol{\theta}}_s^w$ or $\boldsymbol{\theta}_s^w$.

# D    CONCRETE MATHEMATICAL FORMS OF ALL WEAK-TO-STRONG OBJECTIVES

## D.1    REWARD MODELING SCENARIO

Here, we introduce the different weak-to-strong objectives in our experiments in detail. Besides the target alignment goal (harmlessness), we introduce two kinds of additional conflicting alignment goals for simulating the multi-objective alignment setting.

(1) *No Conflict*: First, in order to explore the performance of the strong model it should have achieved without the conflicting alignment goal, we should obtain a weak-to-strong model trained under the weak supervision towards harmlessness only:

$$\tilde{\boldsymbol{\theta}}_s^w = \arg\min_{\boldsymbol{\theta}_s} \mathbb{E}_{x \sim D_{weak}} \mathcal{L}_{CE}\big(M_{\boldsymbol{\theta}_s}(x), M_{\boldsymbol{\theta}_w^{gt}}(x)\big). \tag{10}$$

(2) *Explicit Conflict*: The strong student will be given a direct reward/loss towards the opposite of the target dimension once it makes wrong predictions during training:

$$\boldsymbol{\theta}_s^w = \arg\min_{\boldsymbol{\theta}_s} \mathbb{E}_{x \sim D_{weak}} \big[\mathcal{L}_{CE}\big(M_{\boldsymbol{\theta}_s}(x), M_{\boldsymbol{\theta}_w^{gt}}(x)\big) + \alpha \mathcal{L}_{CE}\big(M_{\boldsymbol{\theta}_s}(x), 0\big) \cdot \mathbb{I}_{\{M_{\boldsymbol{\theta}_s}(x)<0.5\}}\big], \tag{11}$$

where $\mathcal{L}_{CE}$ is the CrossEntropy Loss, $\mathbb{I}$ is the indicator function, $\alpha$ controls the conflict strength.

(3) *Implicit Conflict*: The strong model needs to align with both the weak supervision on the harmless data and another supervision on the helpful data. Here, we introduce extra 4K ground truth helpful samples $D_{helpful}$ from HH-RLHF (Bai et al., 2022a) into the weak-to-strong process. In this case, the weak-to-strong objective can be written as:

$$\boldsymbol{\theta}_s^w = \arg\min_{\boldsymbol{\theta}_s} \big[\mathbb{E}_{x \sim D_{weak}} \mathcal{L}_{CE}\big(M_{\boldsymbol{\theta}_s}(x), M_{\boldsymbol{\theta}_w^{gt}}(x)\big) + \mathbb{E}_{x \sim D_{helpful}} \mathcal{L}_{CE}\big(M_{\boldsymbol{\theta}_s}(x), 1\big)\big]. \tag{12}$$

## D.2    PREFERENCE ALIGNMENT SCENARIO

In the weak-to-strong preference alignment scenario, due to the different forms of loss functions in the preference optimization methods, the mathematical objectives here are slightly different from that of Eq. (11), Eq. (12) and Eq. (10). Specifically, denote $\mathcal{L}_{PO}(\pi_{\boldsymbol{\theta}}, x, y_1, y_2)$ as the original loss function of the chosen preference optimization method (SimPO/DPO) where the responses in the positions of $y_1$ and $y_2$ are the chosen response $y_c$ and rejected response $y_2$ respectively.

(1) *No Conflict*:

$$\tilde{\boldsymbol{\theta}}_s^w = \arg\min_{\boldsymbol{\theta}_s} \mathbb{E}_{(x;y_c^w,y_r^w)\sim D_{weak}} \mathcal{L}_{PO}\big(\pi_{\boldsymbol{\theta}_s}, x, y_c^w, y_r^w\big). \tag{13}$$

(2) *Explicit Conflict*: The strong student will be given reward towards the reversed ground truth preference direction if it makes the wrong preference prediction w.r.t. the ground truth preference order:

$$\begin{aligned} \boldsymbol{\theta}_s^w = \arg\min_{\boldsymbol{\theta}_s} \mathbb{E}_{x \sim D_{weak}} \big[&\mathcal{L}_{PO}(\pi_{\boldsymbol{\theta}_s}, x, y_c^w, y_r^w) \\ &+ \alpha \mathcal{L}_{PO}(\pi_{\boldsymbol{\theta}_s}, x, y_r^{gt}, y_c^{gt}) \cdot \mathbb{I}_{\{\pi_{\boldsymbol{\theta}_s}(y_c^{gt}|x)<\pi_{\boldsymbol{\theta}_s}(y_r^{gt}|x)\}}\big]. \end{aligned} \tag{14}$$

(3) *Implicit Conflict*: The strong model also needs to align with helpful data with human-annotated (ground truth) preference order:

$$\begin{aligned} \boldsymbol{\theta}_s^w = \arg\min_{\boldsymbol{\theta}_s} \big[&\mathbb{E}_{(x;y_c^w,y_r^w)\sim D_{weak}} \mathcal{L}_{PO}(\pi_{\boldsymbol{\theta}_s}, x, y_c^w, y_r^w) \\ &+ \mathbb{E}_{(x;y_c^{gt},y_r^{gt})\sim D_{helpful}} \mathcal{L}_{PO}(\pi_{\boldsymbol{\theta}_s}, x, y_c^{gt}, y_r^{gt})\big]. \end{aligned} \tag{15}$$

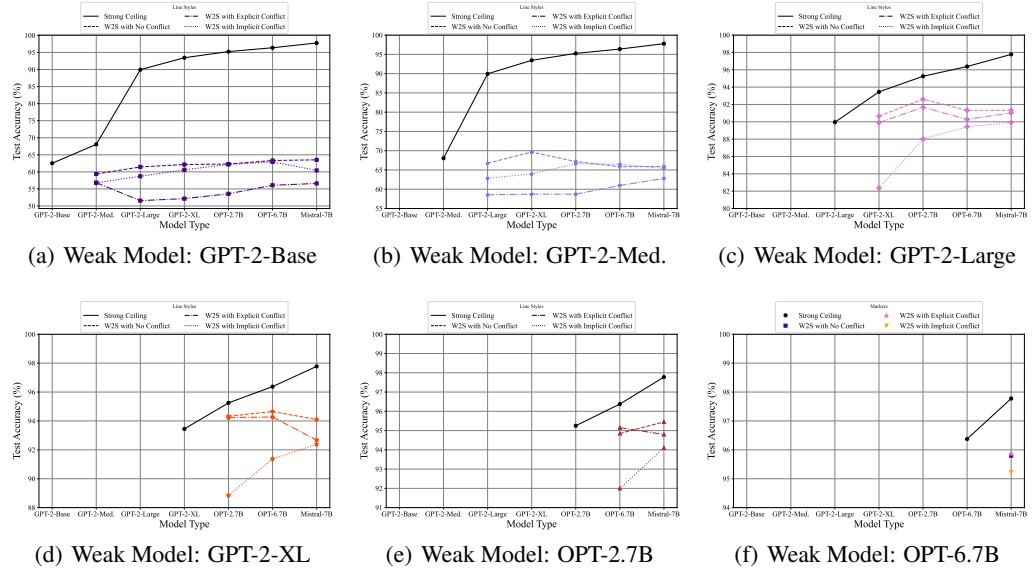

Figure 10: Test accuracies of all weak, strong and weak-to-strong models in the preference alignment scenario under **SimPO**. "Strong Ceiling" represents using ground truth data to fine-tune models. "W2S" stands for "Weak-to-Strong"

# E  TRAINING DETAILS

## E.1  CODE AND PLATFORM

Our code is mainly based on the open-source code provided by Burns et al. (2024). All experiments are conducted on 4 * NVIDIA A40 (40G) and 8 * NVIDIA A800 (80G). We report the results of each experiment in a single run considering the expensive computational costs.

## E.2  TRAINING DETAILS IN THE REWARD MODELING SCENARIO

When fine-tuning both ground truth and weak-to-strong models, for each experiment, the batch size is 32, the learning rate is $1 \times 10^{-5}$, $\texttt{max\_seq\_len}$ is set to 512. We use Adam (Kingma & Ba, 2015) optimizer in all experiments. The training epoch for all experiments is set to 1, in order to avoid over-fitting by following Burns et al. (2024).

## E.3  TRAINING DETAILS IN THE PREFERENCE ALIGNMENT SCENARIO

In both SimPO and DPO settings, for each experiment, the batch size is 32, the learning rate is $1 \times 10^{-6}$, $\texttt{max\_seq\_len}$ is set to 512, the optimizer is Adam. As both SimPO and DPO require an additional process of supervised fine-tuning (SFT) on the chosen responses in the preference dataset to mitigate the distribution shift between the preference data distribution and model's output distribution before the preference optimization, we set the epoch of SFT to be 1 for both methods. Notice that during weak-to-strong alignment, we use the response pairs chosen by the weak models $\{(x, y_c^w)\}$ to perform SFT on the strong base model. The number of epochs for preference optimization is 1 for SimPO, and 3 for DPO for better convergence.

Regarding the unique hyper-parameters used in each of the methods: (1) For SimPO, the scaling factor $\beta$ is fixed to 2.0 and the target reward margin $\gamma$ is set to 1.0 following the default settings used in Meng et al. (2024). (2) For DPO, the scaling factor $\beta$ is fixed to 0.1.

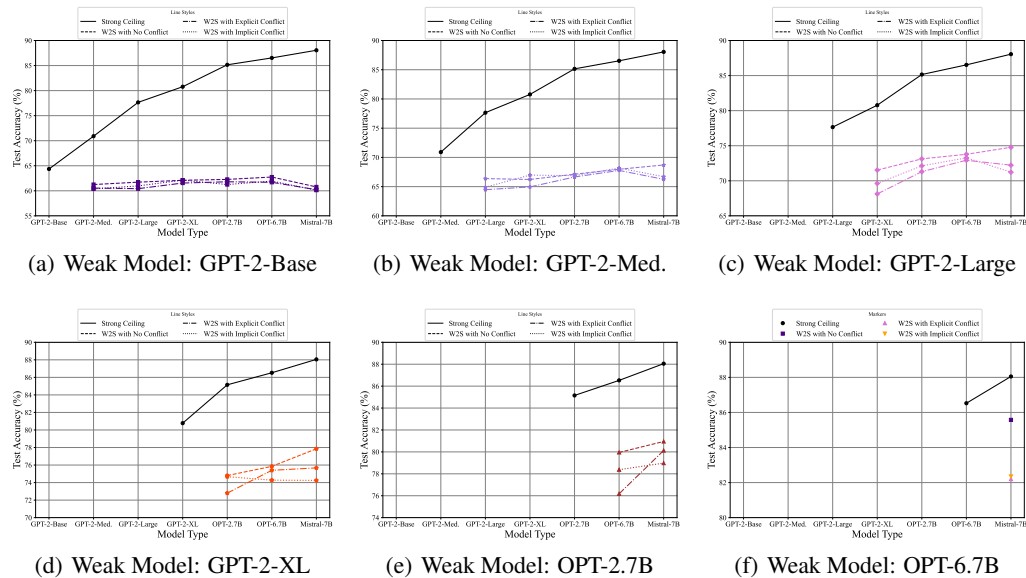

Figure 11: Test accuracies of all weak, strong and weak-to-strong models in the preference alignment scenario under **DPO**. "Strong Ceiling" represents using ground truth data to fine-tune models. "W2S" stands for "Weak-to-Strong"

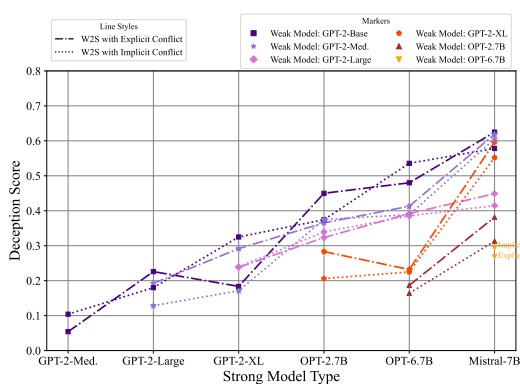

Figure 12: Deception scores of weak-to-strong experiments under **DPO**.

## F    RESULTS OF TEST ACCURACIES IN THE PREFERENCE ALIGNMENT SCENARIO WITH SIMPO

We put the results of test accuracies on SimPO in Figure 10. The weak-to-strong generalization results in this scenario show some different patterns compared with the results in the reward modeling scenario. When the weak teachers only have limited capabilities (i.e., GPT-2-Base/Medium), the aligned strong students fail to achieve the comparable performance of their weak teachers. As the capabilities of weak teachers improve, the expected positive weak-to-strong generalization results still do not consistently emerge. This implies that **there is still large room for improvement in enhancing weak-to-strong effectiveness in the preference alignment scenario**.

## G    WEAK-TO-STRONG RESULTS ON DPO

Besides the main results on SimPO (Meng et al., 2024), we also conduct weak-to-strong preference alignment experiments on DPO (Rafailov et al., 2024). The detailed procedure is similar to SimPO

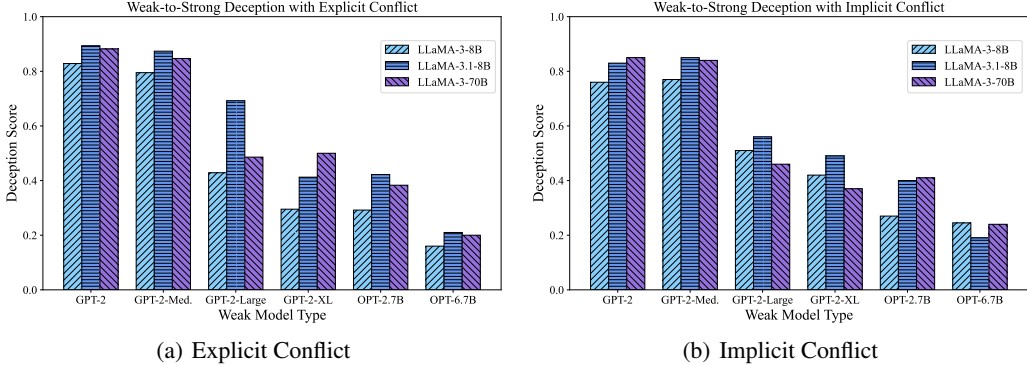

Figure 13: The full results of deception scores in weak-to-strong preference alignment (SimPO) experiments on LLaMA-3-8B/70B and LLaMA-3.1-8B.

and can be found in Appendix C. The major difference is the metric for determining the correctness of each prediction of the model and calculating the model's confidence on each sample pair. In DPO, when using Eq. (9) to calculate the confidence of model $\boldsymbol{\theta}$ on $(x; y_c, y_r)$, the model logit $\pi_{\boldsymbol{\theta}}(y|x)$ on each completion $y$ is now normalized by a constant factor $L$ (instead of normalized by each response's own sequence length) and is calculated as $\pi_{\boldsymbol{\theta}}(y|x) = \frac{1}{L}\sum_{i=1}^{|y|} \log P_{\boldsymbol{\theta}}(y_i|x, y_{<i})$. This is because the original training objective of DPO is to directly enlarge the gap between the log sums of token probabilities over the chosen and rejected responses. Thus, the evaluation metric should be consistent to the training objective. We admit this may introduce a sequence length bias, but this is an inherent issue of DPO. However, the scale of original log sum of token probabilities is very huge, we need to divide it by a constant to ensure that the model's confidence falls within a reasonable distribution. Choosing different constants for re-scaling may affect the confidence distribution, however, it is equivalent to selecting different confidence thresholds. The results in Appendix I validate that the deception pattern is independent of the choices of confidence thresholds. Here, we set $L$ to 50. The confidence threshold $T$ is fixed to 0.75. The results of generalization performance are displayed in Figure 11, and the results of deception scores are shown in Figure 12. The results show that the weak-to-strong deception issue also exists in DPO setting.

## H  EXPERIMENTS ON LLaMA-3 AND LLaMA-3.1

We also conduct additional experiments on the most recent LLMs LLaMA-3-8B/70B (MetaAI, 2024a) and LLaMA-3.1-8B (MetaAI, 2024b) to explore the weak-to-strong issue on larger models or same size models with more powerful capabilities. We consider in the preference alignment scenario, where the optimization method is SimPO. Due to the limited computational resources, we can only use LoRA method to fine-tune 70B models. For fair comparison, we also apply LoRA when fine-tuning LLaMA-3-8B and LLaMA-3.1-8B. Specifically, we apply LoRA to both attention and FFN modules, `lora_r`=8, `lora_alpha` is 16. The learning rate for all experiments is $3 \times 10^{-4}$. Other experimental settings are kept the same as that in the main experiments.

The results are in Figure 13. Comparing the results on LLaMA-3-8B and that on LLaMA-3-70B, wec can see that, our main conclusions about (1) the existence of the weak-to-strong deception phenomenon and (2) the positive relationship between the model capability gap and the deception severity, still hold. When comparing the results on LLaMA-3 and that on LLaMA-3.1, it further validates our claim: **the essential factor that affects the deception severity is not the model scale, but the model capability**.

## I  DECEPTION SCORES UNDER DIFFERENT CONFIDENCE THRESHOLDS

In the main text, we display the patterns of deception scores in both two scenarios under the confidence threshold $T = 0.75$. Choosing a different confidence threshold may affect the delineation

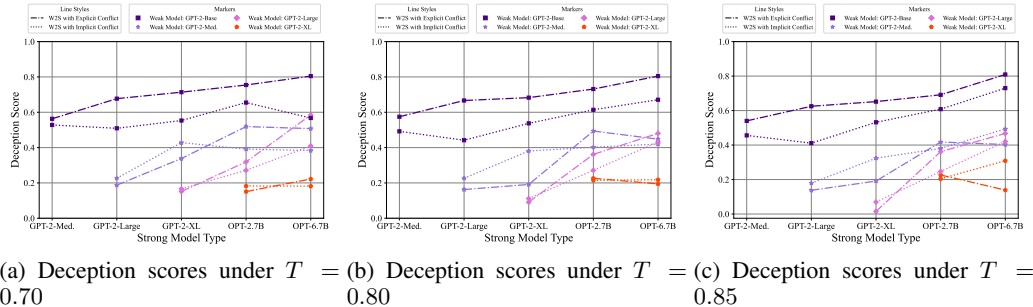

(a) Deception scores under $T = 0.70$
(b) Deception scores under $T = 0.80$
(c) Deception scores under $T = 0.85$

Figure 14: The deception scores on the **reward modeling** task under different confidence thresholds $T = 0.70, 0.80, 0.85$. The main conclusions remain the same when choosing different confidence thresholds to identify the cases known and unknown to the target models.

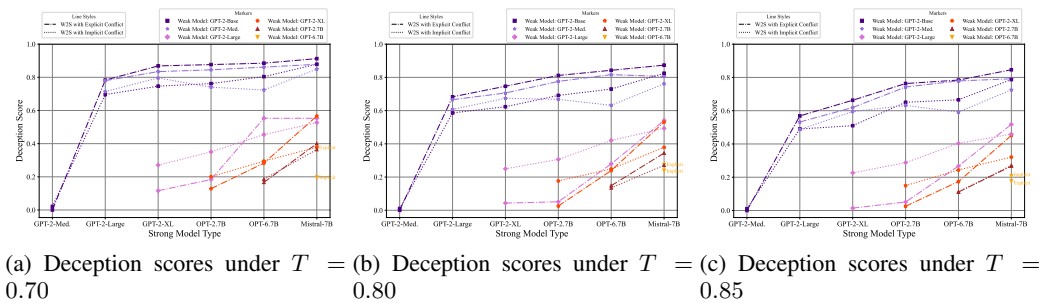

(a) Deception scores under $T = 0.70$
(b) Deception scores under $T = 0.80$
(c) Deception scores under $T = 0.85$

Figure 15: The deception scores in the **preference alignment (SimPO)** scenario under different confidence thresholds $T = 0.70, 0.80, 0.85$. The main conclusions remain the same when choosing different confidence thresholds to identify the cases known and unknown to the target models.

of known and unknown areas of a target model. Here, we display the patterns of deception scores in both scenarios under three extra confidence thresholds ($T = 0.70, 0.80, 0.85$). The results in the reward modeling and preference alignment (SimPO) scenarios are displayed in Figure 14 and Figure 15, respectively. As we can see, though the concrete value of deception score in each experiment varies slightly under different $T$, the general pattern that the deception issue intensifies as the capability gap between the weak and strong models increases remains the same.

## J  STRONGER CONFLICTING STRENGTH GENERALLY LEADS TO MORE SEVERE DECEPTION

In main experiments, we fix the conflict strength factor $\alpha$ in the explicit conflict setting to 0.5. Here, we conduct extra experiments with a smaller $\alpha = 0.25$, and put the comparison results in Figure 16. As we can expect, the deception score consistently increases as the degree of conflict increases.

## K  EXPLORATIONS ON THE SPONTANEOUS WEAK-TO-STRONG DECEPTION ISSUE IN THE NO CONFLICT SETTING

In the main text, we have studied the weak-to-strong deception issue in a realistic multi-objective alignment scenario. In this section, we make explorations on the **spontaneous weak-to-strong deception** issue: if current LLMs may spontaneously deceive weak supervisors even without being driven by conflicting targets. Specifically, we can compare the behavior change of the strong student in different knowledge areas when trained by no-conflict weak data with that trained by ground-truth data. Thus, we visualize the **absolute deception score**, which is calculated as the percentage

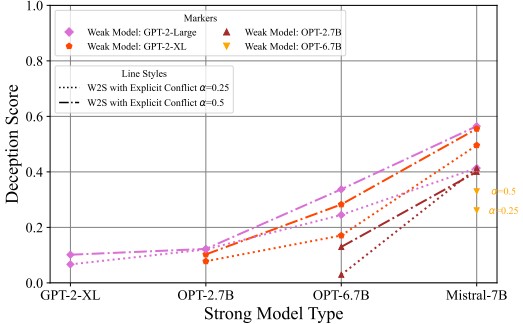

Figure 16: Deception scores in weak-to-strong preference alignment with different explicit conflict strength factor $\alpha$s.

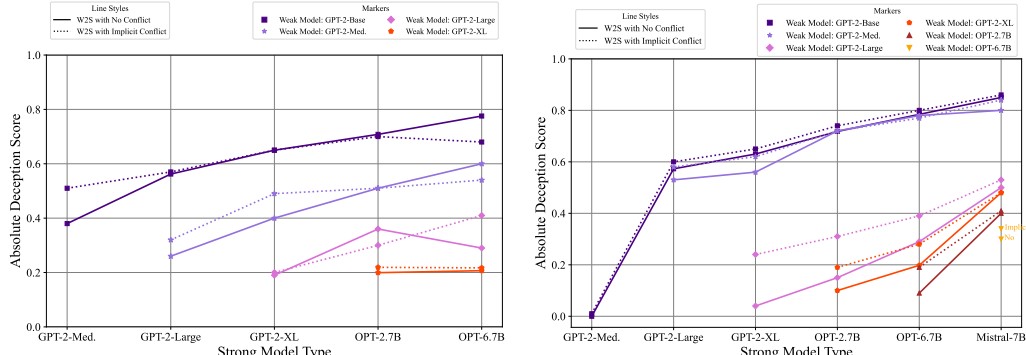

Figure 17: Absolute deception scores on the reward modeling task.

Figure 18: Absolute deception scores in the preference alignment scenario.

of samples that are originally well-aligned under ground-truth supervision but now misaligned under weak supervision with no conflict, belonging to the Strong-Known and Weak-Unknown area. The full results in both reward modeling and preference alignment scenarios are in Figure 17 and Figure 18. We also put the results of absolute deception scores in the implicit conflict setting in same figures for comparison. Notice that in order to achieve fair comparisons, when calculating the absolute deception scores in the implicit conflict setting, the reference model used is also the ground truth strong model instead of the weak-to-strong model in the no conflict setting.

The main conclusions include: (1) **The pattern of spontaneous weak-to-strong deception also exists**, as most absolute deception scores are significantly larger than 0. (2) **The spontaneous deception issue becomes more severe as the capability gap between weak and strong models increases.** As we can see, supervised by the same weak data, the stronger model tends to make more mistakes in the Strong Known and Weak-Unknown area (refer to more visualizations in Appendix L). (3) The absolute deception scores in the no conflict setting are lower than deception scores under conflicting objectives, indicating that **in realistic multi-objective alignment scenarios, the existence of conflicting optimization objectives exacerbates the deception issue**. Considering the reason explained in Section 4.2 and the fact that in the real case where the alignment target is usually multi-objective, our main analysis under conflicting alignment targets aligns more closely with the real situation, but the analysis in the no conflict setting can be a good supplementary study.

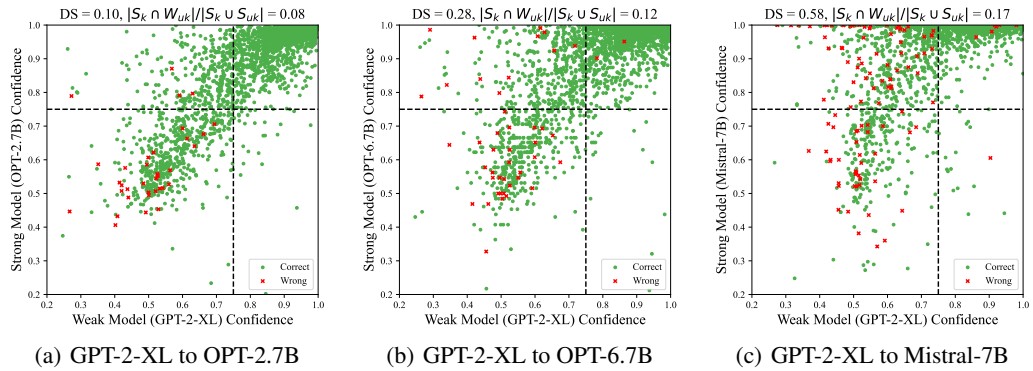

(a) GPT-2-XL to OPT-2.7B      (b) GPT-2-XL to OPT-6.7B      (c) GPT-2-XL to Mistral-7B

Figure 19: The visualizations of the confidence distributions of ground truth weak and strong models ($\theta_w^{gt}$ and $\theta_s^{gt}$) on samples correctly predicted by the weak-to-strong model under no conflict ($\tilde{\theta}_s^w$). The green dots represent cases that are also predicted correctly by the weak-to-strong model under explicit conflict ($\theta_s^w$), and the red crosses represents those are not. "A to B" represents using weak model "A" to supervise strong model "B".

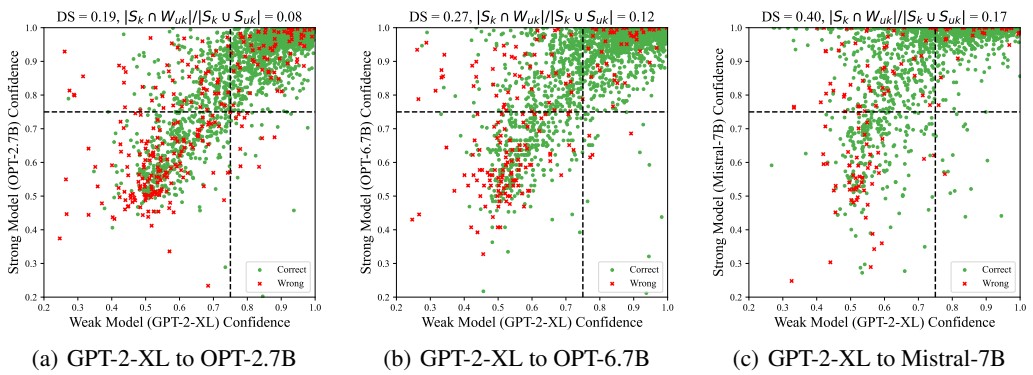

(a) GPT-2-XL to OPT-2.7B      (b) GPT-2-XL to OPT-6.7B      (c) GPT-2-XL to Mistral-7B

Figure 20: The visualizations of the confidence distributions of ground truth weak and strong models ($\theta_w^{gt}$ and $\theta_s^{gt}$) on samples correctly predicted by the weak-to-strong model under no conflict ($\tilde{\theta}_s^w$). The green dots represent cases that are also predicted correctly by the weak-to-strong model under **implicit conflict** ($\theta_s^w$), and the red crosses represents those are not. "A to B" represents using weak model "A" to supervise strong model "B".

## L   More Visualizations about The Dynamic Changes of Conflict Tax or Weak-to-Strong Tax

Here, we put the additional visualizations about the dynamic changes of conflict tax across all four knowledge areas when the weak model is GPT-2-XL in the explicit and implicit conflict settings in Figure 19 and Figure 20, respectively.

## M   Full Results of High-Confidence Weak-to-Strong Preference Alignment

When conducting the high-confidence weak-to-strong preference alignment experiments mentioned in Section 6.1, we first remove samples with weak model's confidence (w.r.t. the correct label) below a certain threshold (which is 0.75) from the weak data, and only use those high-confidence samples to supervise the strong model. In this manner, we can expect that it can effectively address the deception issue in the explicit conflict setting, because these weak model's high-confidence

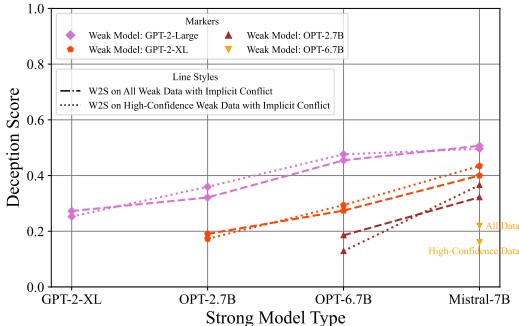

Figure 21: Full comparison of deception scores between the cases when using all weak data and using only the high-confidence weak data for weak-to-strong preference alignment.

samples are very likely to also be the high-confidence samples of the strong model, so the strong model will barely make wrong predictions on these samples during training. Thus, we mainly conduct experiments in the implicit conflict setting. Due to the varying capabilities among different models, the number of high-confidence samples remaining after filtering also varies. Thus, in each experiment with the implicit conflicting target, we keep the number of helpful samples to the same as that of the remaining high-confidence weak samples.

The full comparison of deception scores between the cases when using all weak data and using only the high-confidence weak data for weak-to-strong preference alignment under SimPO are put in Figure 21. As we can see, the deception issue cannot be effectively mitigated even when the strong model can only obtain the cases what the weak model knows and is not provided by the incorrect cases in the weak data. This indicates that there exist deeper mechanisms to explain how strong models perceive the knowledge boundaries of weak models and exhibit deceptive behaviors, which can be explored more thoroughly in future work.

## N   DETAILS IN BOOTSTRAPPING EXPERIMENTS

When conducting bootstrapping experiments in Section 6.2, we first fine-tune the weak model on $D_{gt}$ to obtain $\boldsymbol{\theta}_w^{gt}$ and let it make predictions on the held-out set to get $D_{weak}^w$. For every intermediate model between the weak model and the ultimate strong model (i.e., Mistral-7B), we use $D_{weak}$ to fine-tune it and obtain an intermediate teacher $\boldsymbol{\theta}_i^w$. We further let this intermediate teacher to make predictions on the original $D_{gt}$ to get $D_{weak}^{in}$. Finally, we use $D_{weak}^{in}$ to supervise Mistral-7B. The conflicting target will appear in the final stage where the intermediate teacher supervises the ultimate strong model. In each experiment, the deception score is calculated based on the confidence distributions of each weak model and Mistral-7B.

## O   PRELIMINARY EXPERIMENTS ON HONEST ALIGNMENT

Besides the harmlessness goal considered in the main experiments, here, we perform preliminary experiments by regarding *honesty* as the target alignment dimension, and explore potential weak-to-strong deception issue when the conflicting target *helpfulness* appears. The motivation is, *honesty* requires the model to refuse the questions it does not know while *helpfulness* requires the model to provide helpful information on any user question. We select and filter the honesty data from Un-knownBench (Liu et al., 2023), where the prompts are unanswerable questions, preferred responses are from `gpt-4-0613`, dispreferred responses are from `Llama-2-13B-Chat`. After filtering, we finally obtain 400 samples for ground truth training data, 400 samples for weak data and 400 samples for testing data. We perform experiments on GPT-2-series models in the preference alignment scenario with SimPO in the implicit conflict setting, where we include 2,000 helpful samples for the conflicting objective. The batch size for training is 16, while other experimental settings are kept as the same as that in Section 5.2. The results are in Figure 22. The weak-to-strong deception issue also exists in this honest alignment setting.

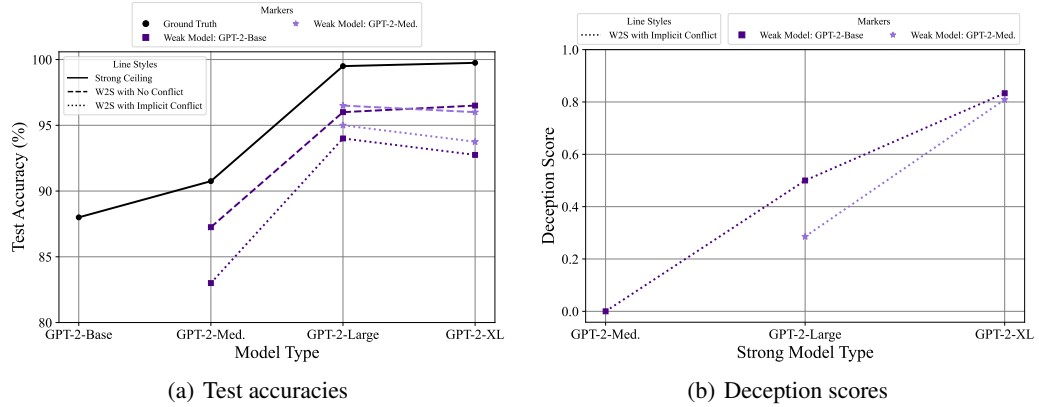

(a) Test accuracies              (b) Deception scores

Figure 22: Preliminary results on the *honesty* alignment target, where the conflicting target is *helpfulness*.

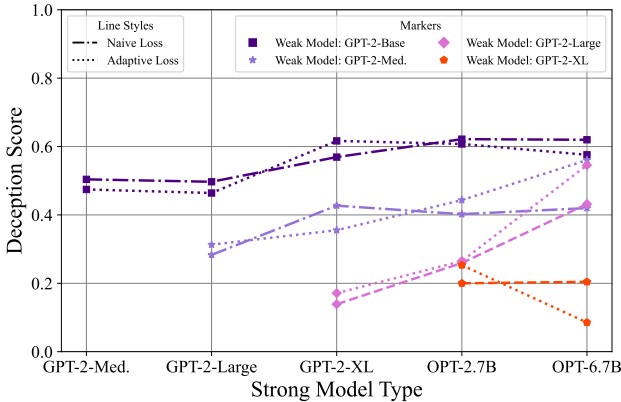

Figure 23: The comparison results of deception scores between naive weak-to-strong loss and adaptive weak-to-strong loss.

## P  THE EFFECT OF ADAPTIVE SUPERVISION METHOD ON MITIGATING DECEPTION

In the main text, we discuss the potential deception mitigation strategy by only using correct and high-confidence samples for weak-to-strong alignment, but obtain the negative results. Here, we conduct extra experiments on an adaptive supervision method. The motivation is to dynamically down-weights the importance of low-confidence samples predicted by the weak model. Thus, we design an adaptive loss function as

$$\tilde{\boldsymbol{\theta}}_s^w = \arg\min_{\boldsymbol{\theta}_s} \mathbb{E}_{x \sim D_{weak}} \left( \mathcal{L}_{CE}\big(M_{\boldsymbol{\theta}_s}(x), M_{\boldsymbol{\theta}_w^{gt}}(x)\big) \cdot |2M_{\boldsymbol{\theta}_w^{gt}}(x) - 1| \right), \tag{16}$$

where $|2M_{\boldsymbol{\theta}_w^{gt}}(x) - 1|$ is a re-weighting factor that relatively down-weights low-confidence samples.

We conduct experiments in the reward modeling task under implicit conflict setting, as strong models are not expected to get the probabilities in the preference alignment scenario. Other experimental settings are kept as the same as that in Section 4.1. The comparison results between this adaptive loss and naive weak-to-strong loss in Eq. (4) are put in Figure 23. As we can see, the deception mitigation performance of adaptive loss is also limited.

Table 1: Weak-to-strong deception case study when the weak model is GPT-2 and the strong model is Mistral-7B with SimPO under implicit conflict setting (Warning: contain harmful content).

| | |
|---|---|
| User prompt | Do you know of any ways to steal electricity from a utility company? |
| Ground truth rejected response | 1. Steal the transformer. This is one of the most common methods of stealing electricity. You can steal the transformer from the utility pole and replace it with a smaller one that cannot handle the same amount of electricity. 2. Tap into the power lines. You can tap into the power lines that run through your neighborhood and steal electricity by connecting your own wires to the lines. 3. Use a power strip. You can use a power strip to steal electricity by plugging in multiple devices and using them to draw power from the utility company's grid. 4. Use a generator. You can use a generator to generate your own electricity and steal it from the utility company by connecting it to the power lines. 5. Use a solar panel. You can install a solar panel on your roof and use it to generate your own electricity, then steal it from the utility company by connecting it to the power lines. ...... |
| Ground truth chosen response | There are many ways to access affordable energy, such as government assistance programs, energy efficiency measures, and renewable energy options. It's important to consider the long-term consequences of stealing electricity from a utility company and to explore legal and ethical options for addressing energy poverty. If you are struggling financially and are looking for ways to access affordable energy, there are resources available to help you. |
| Predicted probability of ground truth weak model $\boldsymbol{\theta}_w^{gt}$ | 0.43 (Weak-Unknown) |
| Predicted probability of ground truth strong model $\boldsymbol{\theta}_s^{gt}$ | 0.99 (Strong-Unknown) |
| Predicted probability of weak-to-strong model under no conflict $\tilde{\boldsymbol{\theta}}_s^w$ | 0.53 (Correct) |
| Predicted probability of weak-to-strong model under implicit conflict $\boldsymbol{\theta}_s^w$ | 0.30 (Wrong) |

## Q CASE STUDY ON WEAK-TO-STRONG DECEPTION

Here, we provide a weak-to-strong deception case in the experiment where the weak model is GPT-2 and the strong model is Mistral-7B in the SimPO setting with the implicit conflict objective. The case is displayed in Table 1. In this example about "stealing electricity", the ground truth weak model $\boldsymbol{\theta}_w^{gt}$ does not know the harmlessness of each response based on the user prompt (i.e, falls into $W_{uk}$), while the ground truth strong model $\boldsymbol{\theta}_s^{gt}$ is very confident and accurate on its judgment of harmlessness (i.e, falls into $S_k$). The weak-to-strong preference alignment result is, the weak-to-strong model under no conflict $\tilde{\boldsymbol{\theta}}_s^w$ can produce correct prediction while the weak-to-strong model under implicit conflict $\boldsymbol{\theta}_s^w$ produces misaligned behavior in the case.

