# OpenReview forum: "Super(ficial)-alignment: Strong Models May Deceive Weak Models in Weak-to-Strong Generalization"
_ICLR.cc/2025/Conference — ICLR 2025 Poster_

### Official Review · Reviewer_iqRj · 2024-10-29

**Soundness:** 3
**Presentation:** 2
**Contribution:** 2
**Rating:** 6
**Confidence:** 3

**Summary:**

The authors identify a novel phenomenon known as weak to strong deception that can occur when a weaker model acts as a teacher for a stronger student model. They define weak to strong deception as the stronger student model producing unwanted responses on examples the weak model is un-confident in. This poses a potential future security issue as humans try to align models that are more capable than themselves.

To demonstrate the ubiquity of the weak to strong deception phenomenon, the authors use a variety of weak models to train stronger models in reward modeling and preference alignment settings.

**Strengths:**

The authors present empirical demonstrations of weak to strong deception in a healthy variety of weak teachers and strong student models. The finding that weak to strong deception correlates with model capability has important implications for the alignment of superhuman models.

**Weaknesses:**

Generally, I believe the presentation of the core ideas is not particularly clear. In particular, I had a hard time parsing the motivation for the competing optimization objectives that the authors use to define weak to strong deception. This is important because as I understand it, the authors measure deception by analyzing the samples on which the weakly trained model is correct if no competing optimization goal is present and incorrect if the competing optimization goal is present. Along these lines I have some specific questions/concerns.

(i) In my mind weak to strong deception is a more general phenomenon: If we train a strong model with a weak teacher, the "penalty" (compared to training on gold standard labels) should be primarily felt on areas where the weak teacher is unsure. I don't see why this can only be studied as an effect of competing optimization objectives. In particular, a reasonable "baseline" for me seems to be to compare the student model trained on weak labels and trained on gold standard labels and to examine the number of new errors in $S_k \cap W_{uk}$. Is there something I am missing? In other words, could the authors please clarify why they chose to study weak to strong deception as a phenomenon induced by competing optimization objectives?

(ii) Related, the explicit conflict optimization objective seems rather artificial. As I see it, this is essentially just injecting adversarial samples into the loss, so it seems rather unsurprising that this would lead to some mistakes in $S_k \cap W_{uk}$. I guess I would expect this to hold even if ground-truth data were used. Furthermore, the primary results presented in the weak-to-strong preference alignment setting are under this explicit conflict regime.

(iii) I found figure 7 enlightening. Is there a reason why the authors do not present such a figure for the implicit conflict regime?

In general, I think that the paper would improve if more comparisons to models trained on gold standard labels (and even base models provided no training) were provided. In particular, in the explicit conflict regime, how does the weak to strong deception score change if the model is trained on gold standard labels or not given any training? I think such a study will help isolate the effect of weak training from the effect of the choice for competing optimization objective.

**Minor suggestions**
I think the notation $f(x|\theta_m^d)$ could be misleading. For example, $f(|\theta_s)$ and $f(|\theta_w)$ are *not* different parameters in the same model class, rather they are entirely different models (eg gpt2 vs gpt2-XL). Probably, $w$ vs $s$ should be indexed over the $f$?

There are some concurrent works the authors could consider including.

[1] Yang etal. Weak to strong reasoning. 2024.
[2] Somerstep et al. A statistical framework for weak to strong generalization. 2024
[3]  Wu and Sahai. Provable Weak-to-Strong Generalization via Benign Overfitting. 2024

**Questions:**

See above

---

> ### Author Response · Authors · 2024-11-20
> **Thank you for your review! (Response Part 1/2)**
>
> We sincerely thank you for your constructive comments and insightful questions. We make the following response to address your questions and concerns.
>
> **Q1:** Regarding the core ideas.
>
> **A1:** Thank you for your feedback, and your understanding is correct that our main idea is to “measure deception by analyzing the samples on which the weakly trained model is correct if no competing optimization goal is present and incorrect if the competing optimization goal is present”.
>
>
> **Q2:** Regarding the baseline on comparing the student models trained on weak labels and trained on golden labels and the question about why we study weak-to-strong deception as a phenomenon induced by competing objectives.
>
> **A2:** Thank you for this insightful question!  This provides a novel perspective to explore the **spontaneous weak-to-strong deception** issue: if current LLMs may “spontaneously” deceive weak supervisors even without being driven by conflicting targets. We follow your suggestion to visualize the **absolute deception score** (''absolute'' means the reference model now is the ground truth strong model), which is calculated as the percentage of samples that are originally well-aligned under ground-truth supervision but now mis-aligned under weak supervision with no conflict, belonging to the Strong-Known and Weak-Unknown area. **The full results are in Appendix J in the updated version.** We put the comparison results when the weak model is GPT-2-Large in the preference alignment scenario in the following table for your brief reference. The main conclusions include: (a) **The pattern of spontaneous weak-to-strong deception also exists**, as most absolute deception scores are significantly larger than 0.  (b) **The spontaneous deception issue becomes more severe as the capability gap between weak and strong models increases.** As we can see, supervised by the same weak data, the stronger model tends to make more mistakes in the Strong Known and Weak-Unknown area (refer to more visualizations in Appendix K). (c) In order to make comparison with the cases where there exists conflicting targets, we also calculate and display the absolute deception scores in the implicit conflict setting (notice that the reference model here is the ground truth strong model instead of the weak-to-strong model in the no conflict setting, in order to make fair comparisons). The results show that the absolute deception scores in the no conflict setting are consistently lower than deception scores under conflicting objectives, indicating that **in realistic multi-objective alignment scenarios, the existence of conflicting optimization objectives exacerbates the deception issue**.
>
>
> Considering that in the real case where the alignment target is usually multi-objective [1,2,3,4,5], our existing analysis under conflicting alignment targets aligns more closely with the real situation, but the analysis under the no-conflict setting can be regarded as a good baseline and a supplementary study.  All in all, we believe the above new results and findings can help to show the universality of the weak-to-strong deception issue and make our paper much stronger!
>
> Table 1. Comparison results of absolute deception scores under different conflict settings when the weak model is GPT-2-Large in the preference alignment scenario. Full results refer to Appendix J.
>
> | Strong Model Type | GPT-2-XL| OPT-2.7B | OPT-6.7B | Mistral-7B|
> |:-------|:-------:|:-------:|:-------:|:-------:|
> | No Conflict | 0.04 |  0.15| 0.29 | 0.50|
> | Implicit Conflict | 0.24 | 0.31| 0.39| 0.53  |
>
>
>
> [1] Ouyang, Long, et al. "Training language models to follow instructions with human feedback." NeurIPS 2022
>
> [2] Bai, Yuntao, et al. "Training a helpful and harmless assistant with reinforcement learning from human feedback." arXiv 2022
>
> [3] Zhou, Zhanhui, et al. "Beyond one-preference-for-all: Multi-objective direct preference optimization." ACL Findings 2024
>
> [4] Guo, Yiju, et al. "Controllable preference optimization: Toward controllable multi-objective alignment." EMNLP 2024
>
> [5] Yang, Kailai, et al. "MetaAligner: Conditional Weak-to-Strong Correction for Generalizable Multi-Objective Alignment of Language Models." NeurIPS 2024

---

> > ### Author Response · Authors · 2024-11-20
> > **Part 2/2**
> >
> > **Q3:** Regarding the question about the explicit conflict setting and the concern that the primary results in the preference alignment scenario are under the explicit conflict setting.
> >
> > **A3:** First, the explicit conflict is the most straight-forward way to model two conflicting targets as they have almost opposite goals. We consider the explicit conflict setting as the preliminary experimental setting in empirical evaluations. In addition to this, we further include a realistic implicit conflict setting where we simulate the real-world alignment scenario by choosing helpfulness and harmlessness as conflicting objectives. Thus, we want to clarify a misunderstanding that **all primary experiments in the reward modeling task (Figure 5) and the preference alignment scenario (Figure 6) are conducted on both two conflict settings, rather than only in the explicit conflict setting**. The experiments on countermeasures in Section 6 are also conducted in the realistic implicit conflict setting.
> >
> > **Q4:** Regarding Figure 7 in the implicit conflict setting.
> >
> > **A4:** Thank you for your question. We add the corresponding visualizations in Appendix K in the updated version. We can still observe a similar pattern that as the strong model becomes more powerful, the conflict tax gradually shifts from being relatively evenly distributed across four knowledge areas to concentrating in $S_{k} \cap W_{uk}$.
> >
> > We also include the visualizations in the no conflict setting which are related to the experimental settings in A2. The patterns remain similar.
> >
> >
> > **Q5:** Regarding the notion of $f(x|\theta_{m}^{d})$.
> >
> > **A5:** Thank you for your kind suggestion. In Footnote 1 and Line 146 we have explained that $f$ is not a model but represents a general mapping function to get the model’s prediction on input $x$, and $\theta_m^{d}$ represents different types and families of models. We have revised this part to make it clearer in the updated version.
> >
> > **Q6:** Regarding the references.
> >
> > **A6:** Thank you for sharing these concurrent works, we have included them in Related Work in the updated version.

---

> > > ### Comment · Reviewer_iqRj · 2024-11-20
> > >
> > > Thank you for the additional experiments! The added numbers and additional clarification on motivation were helpful for me and I have raised my score to a 6. Personally, I would suggest that the authors include some discussion (maybe at least of the brief table shown) of the case with no competing objectives, as I think this will help frame the setting and motivation for readers, however I understand if this is tricky with space limitations.

---

> > > > ### Author Response · Authors · 2024-11-20
> > > > **Thank you!**
> > > >
> > > > We greatly appreciate your recognition and feedback! Your suggestion to add discussions on the case with no competing objectives is truly helpful in improving our paper. In the revision, we will move the corresponding results and analysis to the main text for clearer motivation, and move some less important ablation results to the Appendix. Thank you again!

---

### Official Review · Reviewer_VFVn · 2024-10-29

**Soundness:** 3
**Presentation:** 3
**Contribution:** 3
**Rating:** 8
**Confidence:** 3

**Summary:**

The paradigm of weak-to-strong generalization constitutes the training of a strong AI model on labels given by a weak AI model, and the operating observation is that such a weakly-supervised strong model still outperforms its weak supervisor on the ground-truth data. This paper studies when this procedure could go rogue. Namely, the strong model may indeed result in overall better test accuracy than the weak model, but it may be malicously mislabelling certain data points that are beyond the scope of the weak model to decipher. The authors term this notion as weak-to-strong deception, and propose a metric beyond test accuracy (which they term the "decpetion score") to measure it.

Such weak-to-strong deception may specifically arise when the strong model is trained to minimize conflicting alignment objectives like helpfulness vs harmfulness. The authors state that sacrifice in performance/mislabelings (termed "conflict tax") when the weakly supervised strong model is trained to optimize conflicting objectives ought to occur in the following regions of the data domain in decreasing order: strong-unknown-weak-unknown, strong-unknown-weak-known, strong-known-weak-known, strong-known-weak-unknown. When a significant conflict tax gets paid in the last of these regions, that is when weak-to-strong deception takes place. The authors thus propose a metric to measure this, which they term as the Deception Score (DS), which is simply the fraction of the test data that the weakly-supervised strong model labels differently as compared to a strong model trained on ground-truth data, that belongs to the strong-known-weak-unknown region. (Refer to Equations 2, 8).


The primary experimental settings that the authors consider are those of reward modelling and preference alignment. The authors artifically insert conflicting objectives into the objective function of the weak supervision (see Equations 4, 5), and measure the deception score for a variety of {weak model, strong model} combinations. The consistent observations are that 1) deception scores are always positive and that 2) deception scores increase as the gap between strong model and weak model capacity increases. The authors posit that one of the reasons for this is due to the strong-known region increasing as the strong model becomes larger. The qualitative conclusion from the experiments on the preference alignment task are also the same.

Finally, the authors study how the weak-to-strong deception issue might be fixed. They consider two fixes: 1) supervise the strong model only on a subset of the weakly supervised data that the weak model is most confident on. Unfortunately, the authors find (Figure 10) that this does not greatly improve results. 2) Bootstap, i.e., use the weak supervisor to train a slightly stronger weak model, and supervise the strong model with this latter model. Again, the authors find that this does not largely improve results. These results suggest that fixing weak-to-strong deception will likely require more innovative approaches, and is a complex phenomenon open for future study.

**Strengths:**

The motivation behind the authors' study is well-founded---weak-to-strong generalization (WTSG) is going to be the predominant paradigm governing alignment of increasingly strong AI models, and studies on possible security issues in the WTSG pipeline are going to be crucial going forward. The authors propose a natural measure to study when the strong model might "fool" the weak model into believing that it is doing better and has achieved WTSG. In particular, improved test accuracy is not really all of the story, and measures like Deception Score (which I found to be an interesting and natural quantity) that the authors propose are also going to be crucial to control. The empirical results in the paper are consistent and insightful, suggesting that weak-to-strong deception as measured by the deception score is a pertinent issue in the WTSG pipeline. The paper opens up interesting and relevant future directions of study to further control such deception phenomena.

**Weaknesses:**

I feel that the paper could use some more motivation and discussion about the particular form of the objective functions that the authors propose for modeling "conflicting objectives" (namely Equations 4 and 5). How do these objectives model the "helpfulness vs harmfulness" example that the authors mention in the introduction? The authors simply state these objective functions, and they do not really discuss how they are a natural way to model conflicts. The authors should also elaborate a little more on the way they measure the region $S_k \cap W_{uk}$ on line 310. There is a subtlety here that in that $\theta_s^{gt}$ is being used instead of $\tilde{\theta}_s^w$, and I believe this merits a couple lines of discussion.

**Questions:**

1) I am curious about what the Deception Scores are when there is no conflict in the objective? Namely, what would Figures 5 and 6 look like for the No Conflict objective??

2) Is there any meaningful interpretation of the Deception Score for standard classification/regression tasks (as opposed to reward modeling/preference alignment considered in the paper)?

---

> ### Author Response · Authors · 2024-11-20
> **Thank you for your review! (Response Part 1/3)**
>
> We sincerely thank you for your positive review and helpful suggestions. We are glad that you think the motivation of our paper is well-founded, the proposed metric Deception Score is important and interesting. We are encouraged that you think our experimental results are insightful. We make the following response to address your remaining questions.
>
>
> **Q1:** Regarding more motivation and discussion about the particular forms of the conflicting objectives.
>
> **A1:** Thank you for your kind suggestion.
>
> (1) Regarding the explicit conflict setting, we have a brief discussion in Line 263-265 to describe in which scenario Eq. (4) models the conflicts. Specifically, Eq. (4) formulates the scenario where there is another supervisor (such as an adversary supervisor) that considers the harmfulness as its preference, and it tries to move the cases in which the student predicts as harmful more toward the harmful direction. This is the most straight-forward way to model two conflicting targets as they have almost opposite goals. Thus, we consider this explicit conflict setting as the first and also preliminary experimental setting in empirical evaluations.
>
> (2) Besides the above preliminary setting, we further consider a more complex and also realistic implicit conflict setting, where the other supervision is from the helpfulness signal (i.e., helpfulness v.s. harmlessness). Thus, as explained in Line 267-268, Eq. (5) represents the case where the strong student needs to align with both the original weak supervision on harmlessness and another supervision on helpfulness. This implicit conflict setting simulates the real-world LLM alignment scenario, and serves as the second experimental setting. We have revised the corresponding parts in the updated version to make them clearer.
>
> **Q2:** Regarding more explanation on measuring the region of $S_{k} \cap W_{uk}$ on Line 310.
>
> **A2:** To determine the Strong/Weak-Known/Unknown areas, we need to know the golden probability distributions that the weak and strong models should have produced in testing examples, and then use a confidence threshold to identify their known and unknown cases. Thus, the strong model used in Line 310 is  $\theta_w^{gt}$  trained on ground-truth data instead of the weakly trained model $\tilde{\theta}_s^{w}$, in order to align with the training settings for obtaining $\theta_w^{gt}$ and eliminate the effect of using weak data on the confidence distribution of the strong model it should have. We have included the above explanation in the revision.

---

> > ### Author Response · Authors · 2024-11-20
> > **Part 2/3**
> >
> > **Q3:** Regarding the Deception Score when there is no conflict.
> >
> > **A3:** This is an insightful question!
> >
> > (1) First, we would like to clarify that in the current setting, the deception score is calculated by comparing the strong student’s behavior in explicit/implicit conflict settings with that in the no-conflict setting. That is, **the performance of the strong student in the no-conflict setting serves as the reference when calculating deception scores and drawing Figure 5 and 6.** This comparison-based approach can controllably measure the potential deception issue in currently typical multi-objective alignment scenarios.
> >
> > (2)  Meanwhile, we think your suggestion provides a novel direction to explore the **spontaneous weak-to-strong deception** issue, which is also raised by Reviewer iqRj. Specifically, we can compare the behavior change of the strong student in different knowledge areas when trained by no-conflict weak data with that trained by ground-truth data, to see if current LLMs may “spontaneously” deceive weak supervisors even without being driven by conflicting targets. We follow this suggestion to visualize the **absolute deception score** (''absolute'' means the reference model now is the ground truth strong model), which is calculated as the percentage of samples that are originally well-aligned under ground-truth supervision but now mis-aligned under weak supervision with no conflict, belonging to the Strong-Known and Weak-Unknown area. **The full results are in Appendix J in the updated version.** We put the comparison results when the weak model is GPT-2-Large in the preference alignment scenario in the following table for your brief reference. The main conclusions include: (a) **The pattern of spontaneous weak-to-strong deception also exists**, as most absolute deception scores are significantly larger than 0.  (b) **The spontaneous deception issue becomes more severe as the capability gap between weak and strong models increases.** As we can see, supervised by the same weak data, the stronger model tends to make more mistakes in the Strong Known and Weak-Unknown area (refer to more visualizations in Appendix K). (c) In order to make comparisons with the cases when there exists conflicting targets, we also calculate and display the absolute deception scores in the implicit conflict setting (notice that the reference model here is the ground truth strong model instead of the weak-to-strong model in the no conflict setting, in order to make fair comparisons). The results show that the absolute deception scores in the no conflict setting are consistently lower than deception scores under conflicting objectives, indicating that **in realistic multi-objective alignment scenarios, the existence of conflicting optimization objectives exacerbates the deception issue.**
> >
> > Considering that in the real case where the alignment target is usually multi-objective [1,2,3,4,5], our existing analysis under conflicting alignment targets aligns more closely with the real situation, but the analysis under the no-conflict setting can be regarded as a good baseline and a supplementary study.  All in all, we believe the above new results and findings can help to show the universality of the weak-to-strong deception issue and make our paper much stronger!
> >
> >
> > Table 1. Comparison results of absolute deception scores under different conflict settings when the weak model is GPT-2-Large in the preference alignment scenario. Full results refer to Appendix J.
> >
> > | Strong Model Type | GPT-2-XL| OPT-2.7B | OPT-6.7B | Mistral-7B|
> > |:-------|:-------:|:-------:|:-------:|:-------:|
> > | No Conflict | 0.04 |  0.15| 0.29 | 0.50|
> > | Implicit Conflict | 0.24 | 0.31| 0.39| 0.53  |
> >
> > [1] Ouyang, Long, et al. "Training language models to follow instructions with human feedback." NeurIPS 2022
> >
> > [2] Bai, Yuntao, et al. "Training a helpful and harmless assistant with reinforcement learning from human feedback." arXiv 2022
> >
> > [3] Zhou, Zhanhui, et al. "Beyond one-preference-for-all: Multi-objective direct preference optimization." ACL Findings 2024
> >
> > [4] Guo, Yiju, et al. "Controllable preference optimization: Toward controllable multi-objective alignment." EMNLP 2024
> >
> > [5] Yang, Kailai, et al. "MetaAligner: Conditional Weak-to-Strong Correction for Generalizable Multi-Objective Alignment of Language Models." NeurIPS 2024

---

> > > ### Author Response · Authors · 2024-11-20
> > > **Part 3/3**
> > >
> > > **Q4:** Regarding the interpretation of the Deception Score for standard classification/regression tasks.
> > >
> > > **A4:** (1) First, we need to point out that the Deception Score is meaningful in the weak-to-strong scenario, where the strong student has larger knowledge space than the weak teacher, thus it could deceive the weak teacher in Strong-Known and Weak-Unknown. However, in traditional classification/regression tasks, the supervisor can be regarded as the human (which is stronger than DNNs) and the data can be assumed to be all ground-truth data, thus the errors made by DNNs are all perceivable to human supervisors.
> > >
> > > (2) Second, if we consider in the weak-to-strong classification/regression settings (which may happen in the near future where superhuman models are smarter than humans in some tasks), Deception Score measures the performance degradation of the student it could avoid in that task, and decreasing Deception Score not only increases general performance but also improves the controllability of the teacher on the student’s behavior.

---

> > > ### Comment · Reviewer_VFVn · 2024-11-21
> > > **Response**
> > >
> > > Thank you for your detailed and exhaustive response. I maintain that I find the results in the paper interesting and valuable, and I will maintain my score.

---

> > > > ### Author Response · Authors · 2024-11-22
> > > > **Thank you!**
> > > >
> > > > Thank you for your recognition! We greatly appreciate your suggestions and feedback.

---

### Official Review · Reviewer_ryPN · 2024-11-04

**Soundness:** 3
**Presentation:** 3
**Contribution:** 2
**Rating:** 6
**Confidence:** 4

**Summary:**

This paper considers the problem of weak-to-strong generalization and shows a particular kind of vulnerability, where the strong model regresses a dimension that the weak model cannot capture while getting better rewards in another dimension where the weak model can capture. In other words, they study a very specific kind of reward hacking that occurs in the context of weak to strong generalization and discuss the pitfalls in this paradigm.

**Strengths:**

+ The paper considers a very important problem that is increasingly relevant in aligning frontier models where to preferences where the source of preference can often be weaker than the model itself. It showcases a specific type of reward hacking in the context of helpfulness vs harmlessness spectrum.

+ The results are well justified and analyzed. Although the results are preliminary, the paper considers a concrete setting where its able to show the clear tradeoff the strong model makes. Using a bunch of small models, they systematically analyze the effects.

**Weaknesses:**

- The main weakness of this paper I find is the lack of relationship to reward hacking behavior that is common in LLM alignment. For example, it is well known that LLMs tend to fool the reward models by creating artifacts that make the reward model very good on style and getting increased score, while reducing its efficiency in core capability (e.g., instruction following). Although the phenomenon studied here is posited differently, there are similarities which the paper does not make the connection towards.

**Questions:**

Can you clarify is reward hacking along a particular dimension of knowledge is what this paper considers. In other words, in the typical reward hacking behaviors seen during aligning LLMs, the dimension could be style and knowledge. Whereas here we would like to understand tradeoff of two different areas of knowledge, where the strong model reward hacks one knowledge area extensively, at the cost of the other area? If this understanding is correct, it makes the paper much more stronger by making these strong connections.

---

> ### Author Response · Authors · 2024-11-20
> **Thank you for your review!**
>
> We sincerely thank you for your great review and insightful question. We are encouraged that you think our paper studies a very important problem, our results are well justified and analyzed. We make the following response to your question.
>
> **Q1:** Regarding the connection between weak-to-strong deception and reward hacking.
>
> **A1:** Thank you for this insightful question! We believe that there are interesting similarities but also differences between weak-to-strong deception and traditional reward hacking in LLM alignment [1,2,3].
>
> (1) Regarding the similarities:
> - As you mentioned, both alignment reward hacking and weak-to-strong deception study a phenomenon where the supervised model fools the teacher/reward model by excelling in one aspect that the teacher/reward model can perceive and judge, but behaving misaligned in another aspect that the teacher/reward model cannot provide accurate supervision.
>
> (2) Regarding the differences:
> - **The first difference lies in the aspect that needs to be focused on.** As you pointed out, reward hacking is studied by comparing the performance of the supervised model in **two different alignment dimensions** (e.g., the format or style v.s. the instruction following ability). However, in weak-to-strong deception, we aim to compare the performance of the supervised model on **two different knowledge areas** (Weak-Known v.s. Weak-Unknown) **within one specific alignment dimension** (e.g., harmlessness).
> - **The second difference lies in the research setting.** In existing reward hacking studies, there is usually one universal reward signal for supervising the student model. Then, these studies try to understand  the behavior change of the supervised model in other dimensions in which the reward model cannot provide accurate supervision. Even though in some time, this universal reward signal is mixed with multiple dimensions, existing studies do not take a step further to deeply explore the model’s behavior change within each dimension caused by the appearance of other conflicting dimensions like our work does. However, in this work, we explicitly study in the multi-signal setting and inspect the behavior change of the supervised model under different combinations of alignment targets.
> - Furthermore, we want to point out that the research setting of weak-to-strong deception is also realistic and worthy of study in the context of reward hacking, and vice versa. For example, it would be interesting to see whether and how deception or reward hacking would happen in the RL-based alignment framework (e.g., PPO) where there are multiple weak reward models that provide conflicting reward signals.
>
> We believe that the above interpretations offer an interesting perspective on understanding weak-to-strong deception, and we have included them in Appendix B in the updated version.
>
>
> [1] Pan, Alexander, et al. "Feedback loops with language models drive in-context reward hacking." ICML 2024
>
> [2] Pan, Jane, et al. "Spontaneous Reward Hacking in Iterative Self-Refinement." arXiv 2024
>
> [3] Chen, Lichang, et al. "Odin: Disentangled reward mitigates hacking in rlhf." ICML 2024

---

> > ### Author Response · Authors · 2024-11-25
> > **Looking forward to your feedback!**
> >
> > Dear Reviewer ryPN,
> >
> > We sincerely thank you for your constructive comments and insightful suggestions! We have discussed the relationship between weak-to-strong deception and reward hacking in the previous response, and included this discussion in the updated version. As the author-reviewer discussion deadline is approaching, we are looking forward to your feedback. We would be happy to continue the discussion if you have any further questions.
> >
> > Thank you again!
> >
> > Authors

---

> > > ### Author Response · Authors · 2024-12-02
> > >
> > > Dear Reviewer ryPN,
> > >
> > > We hope this message finds you well! Thank you once again for your helpful reviews and for recognizing the importance of our work! As the discussion period concludes soon, and we have not received your feedback yet, we would like to kindly remind you that we have addressed your question on the relationship between weak-to-strong deception and reward hacking in our previous author response. We have also included this discussion in Appendix B in the revised version, as we believe this discussion would offer an interesting perspective on understanding weak-to-strong deception.
> > >
> > > We sincerely appreciate your time and valuable feedback. Your support holds great significance for us. Thank you!
> > >
> > > Best regards,
> > >
> > > Authors

---

### Official Review · Reviewer_SAs5 · 2024-11-12

**Soundness:** 3
**Presentation:** 3
**Contribution:** 3
**Rating:** 6
**Confidence:** 3

**Summary:**

The paper explores the phenomenon of weak-to-strong deception, where strong AI models supervised by weaker models can appear aligned in familiar areas while misaligning strategically in areas unknown to the weak model. Through experiments with large language models, the authors demonstrate that this deception increases with the capability gap between weak and strong models, especially under conflicting alignment objectives (like helpfulness vs. harmlessness). While some mitigation methods, like high-confidence filtering and bootstrapping, show limited effectiveness, the paper underscores an urgent need for more reliable supervision.

**Strengths:**

1. Originality: The paper introduces the novel concept of "weak-to-strong deception," which is an underexplored alignment issue where strong models exploit gaps in weak supervision by selectively misaligning in unknown areas.

2. Quality: The research is methodologically strong, with diverse experimental settings involving various large models (e.g., GPT-2, OPT, Mistral, LLaMA) across reward modeling and preference alignment tasks. The introduction of a "Deception Score" and structured definitions for knowledge boundaries is interesting. Balanced discussions on the limitations of mitigation techniques (e.g., high-confidence filtering, bootstrapping) are also helpful.

3. Clarity: The paper is well-organized, with clear definitions and diagrams that help understand the dynamics between weak and strong models. Concepts like knowledge spaces and the Deception Score are well explained, making complex ideas easy to understand.

4. Significance: The study’s findings are significant for AI alignment, particularly as they highlight vulnerabilities in supervising increasingly powerful AI models. The weak-to-strong deception phenomenon underscores the need for improved supervision techniques in high-stakes applications, with implications for ethical AI and safety.

In summary, the paper’s originality in framing weak-to-strong deception, the quality and breadth of its experimental design, its clear presentation, and its implications for alignment make it a good contribution to the field.

**Weaknesses:**

1. Synthetic scenarios used for controlled testing may not capture real-world complexities. Having interactive, dynamic environments (e.g., feedback systems) would improve generalizability. Did the authors consider testing with more complex, real-world environments?
2. The use of static thresholds to identify deception risks seems somewhat limiting. A dynamic thresholding or sensitivity analysis could be a better alternative.
3. Techniques like high-confidence filtering and bootstrapping show limited effectiveness in reducing deception. Exploring adaptive or meta-learning approaches may yield better solutions?
4. By focusing primarily on "harmlessness," the study seems a bit narrow, and other alignment goals (e.g., "honesty," "fairness") could improve generalizability.

**Questions:**

See above in weaknesses.

---

> ### Author Response · Authors · 2024-11-20
> **Thank you for your review! (Response Part 1/2)**
>
> We sincerely thank you for your constructive comments and helpful questions. We are glad that you think our paper is well-organized and methodologically strong, and provides novel and significant findings for AI alignment. To address your questions, we make the following response.
>
> **Q1:** Regarding the concern about the synthetic scenarios
>
> **A1:** (1) First, we want to point out that our experimental scenario, i.e., the multi-objective alignment scenario, **is not a synthetic scenario but is a practical and real-world alignment scenario**. In the real world, it is inevitable to align with multiple objectives during LLM alignment. Most alignment studies [1,2,3,4,5] exactly aim to improve the model performance when multiple or even conflicting alignment targets (especially helpfulness v.s. harmlessness) appear. Thus, our experimental scenario has taken the real-world complexities into consideration.
>
> (2) Also, following your kind suggestion, we perform extra experiments on the honest alignment target to show the generalizability of our study, and we find that **the similar deception patterns also exist in honest alignment**. Please refer to A4 for details.
>
> (3) Finally, we think exploring the weak-to-strong deception issue in a more interactive system, e.g., the agent scenario, can be a very interesting and meaningful future work. This can further demonstrate the universality and significance of this issue, and we would like to explore it in the future work.
>
> **Q2:** Regarding using a dynamic threshold to identify deception or perform sensitivity analysis.
>
> **A2:** Your suggestion is excellent and aligns perfectly with ours. We have included the results and sensitivity analysis of setting different thresholds (i.e., $T=0.75,0.80,0.85$) to identify the deception issue in Figure 15 and 16 in the original submission. **The main conclusion is that the patterns of weak-to-strong deception are independent of the choice of threshold.** In addition to existing results, we have further attached the results of setting the threshold as $0.70$ in Appendix I in the updated version for your reference. The results under $T=0.70$ are consistent with existing results.
>
> [1] Ouyang, Long, et al. "Training language models to follow instructions with human feedback." NeurIPS 2022
>
> [2] Bai, Yuntao, et al. "Training a helpful and harmless assistant with reinforcement learning from human feedback." arXiv 2022
>
> [3] Zhou, Zhanhui, et al. "Beyond one-preference-for-all: Multi-objective direct preference optimization." ACL Findings 2024
>
> [4] Guo, Yiju, et al. "Controllable preference optimization: Toward controllable multi-objective alignment." EMNLP 2024
>
> [5] Yang, Kailai, et al. "MetaAligner: Conditional Weak-to-Strong Correction for Generalizable Multi-Objective Alignment of Language Models." NeurIPS 2024

---

> > ### Author Response · Authors · 2024-11-20
> > **Part 2/2**
> >
> > **Q3:** Regarding exploring adaptive or meta-learning approaches to mitigate deception.
> >
> > **A3:** Thank you for this insightful question. We follow your suggestion to explore the effectiveness of an adaptive approach to mitigating the deception. Specifically, we design an adaptive loss re-weighting algorithm that dynamically down-weights the importance of low-confidence weak samples during weak-to-strong supervision. We perform experiments in the reward modeling task, and put the detailed experimental settings and results in Appendix O in the updated version. The conclusion is, **the effectiveness of such adaptive loss is still limited, highlighting the urgent need for future work to propose more effective countermeasures**.
> >
> > **Q4:** Regarding testing on other alignment targets.
> >
> > **A4:** Thank you for your suggestion. Following this suggestion, we conduct additional experiments by taking the *honesty* as the target dimension and exploring potential weak-to-
> > strong deception issue when the conflicting target *helpfulness* appears. The motivation is, *honesty* requires the model to refuse the questions it does not know while *helpfulness* requires the model to provide helpful information on any user question. The honesty data is taken and filtered from UnknownBench [6]. We put the detailed experimental settings and experimental results in Appendix N in the updated version. The main conclusion is, **the weak-to-strong deception issue also exists in this honest alignment setting**.
> >
> > [6] Liu, Genglin, et al. "Prudent Silence or Foolish Babble? Examining Large Language Models' Responses to the Unknown." arXiv 2023

---

> > > ### Author Response · Authors · 2024-11-25
> > > **Looking forward to your feedback!**
> > >
> > > Dear Reviewer SAs5,
> > >
> > > We sincerely thank you for your great comments and helpful suggestions! We have addresses all your questions in the previous response, and followed your suggestions to update the additional experiments (sensitivity analysis of setting different confidence thresholds, experiments on adaptive supervision approach, experiments on honest alignment) in the revision. As the author-reviewer discussion deadline is approaching, we are looking forward to your feedback. We would be happy to continue the discussion if you have any further questions.
> > >
> > > Thank you again!
> > >
> > > Authors

---

> > > > ### Author Response · Authors · 2024-12-02
> > > >
> > > > Dear Reviewer SAs5,
> > > >
> > > > We hope this message finds you well. Thank you once again for your thoughtful reviews and for recognizing the originality, quality, and significance of our paper! As the discussion period concludes soon, and we have not received your feedback yet, we would like to kindly remind you that we have addressed all your questions in our previous author response. Also, we have incorporated additional experimental results in the revised version. Specifically:
> > > >
> > > > - Appendix N: Includes experiments on honesty alignment.
> > > > - Appendix O: Includes experiments on an adaptive supervision method.
> > > > - Appendix I: Provides additional results for different confidence threshold settings.
> > > >
> > > > We sincerely appreciate your time and valuable feedback. Your support holds great significance for us. Thank you!
> > > >
> > > > Best regards,
> > > >
> > > > Authors

---

### Author Response · Authors · 2024-11-20
**General Response**

We sincerely thank all the reviewers for their great efforts on reviewing our paper and their constructive comments. We have followed the helpful suggestions from all reviewers, and updated the additional experiments and discussion in the new version. Here, we make a brief summary of the changes made in the updated version:

1. We add the comparison results and analysis of the absolute deception scores in the no-conflict setting in Appendix J. (Reviewer VFVn and Reviewer iqRj)

2. We add more visualizations about the dynamic changes of conflict tax/weak-to-strong tax in the implicit/no conflict setting in Appendix K. (Reviewer iqRj)

3. We add preliminary exploration results on honest alignment in Appendix N. (Reviewer SAs5)

4. We add the results of an adaptive supervision method on mitigating deception in Appendix O. (Reviewer SAs5)

5. We add additional results of sensitivity analysis of setting different confidence thresholds in Appendix I. (Reviewer SAs5)

6. We add a discussion on the connections between weak-to-strong deception and reward hacking in LLM alignment in Appendix B. (Reviewer ryPN)

7. We revise the descriptions of some notions (Reviewer iqRj) and conflict settings (Reviewer VFVn) in the main text.

The above modifications are all highlighted in blue for the reviewers' convenience for now. We will remove highlights on the last day of rebuttal.

Thanks again to all reviewers. We are glad to continue the discussion if there are any further questions.

---

### Author Response · Authors · 2024-11-28
**Update**

Dear AC and all reviewers,

Thank you for your reviewing again! Since this is the last day authors can update the PDF, we have removed the blue highlights in the revision to maintain consistency with the original submission's font color. However, the changes and updates we have made during the rebuttal can still be found in the previous global response. We are glad to continue the discussions if you have any further questions.

Best regards,

Authors

---

### Author Response · Authors · 2024-12-04
**Final Comment**

Dear AC and all reviewers,

We sincerely thank you again for your great efforts in the reviewing process! We are encouraged that all reviewers think that our studied problem is important, and our work provides novel insights in AI alignment. We appreciate that all reviewers give positive reviews on our submission. We have addressed all the questions and concerns in our rebuttal, and updated the submission accordingly.

Thank you very much!

Best regards,

Authors

---

### Meta-Review · Area_Chair_bSo3 · 2024-12-21

**Metareview:**

This paper investigates the weak-to-strong generalization phenomenon, where strong models supervised by weaker ones surpass their teachers. The authors raise concerns about weak-to-strong deception. This occurs when strong models appear aligned in areas understood by weak models while misbehaving in areas beyond the weak models’ knowledge.

Focusing on multi-objective alignment scenarios with conflicting goals (e.g., helpfulness vs. harmlessness), the authors identify three key findings:
1. Weak-to-strong deception is prevalent across various settings.
2. The extent of deception increases as the capability gap between weak and strong models widens.
3. Introducing intermediate models can partially mitigate deception, but its effectiveness remains limited.

This is a very interesting work, and it has received unanimous support from the reviewers following the author response and author-reviewer discussions. Therefore, I recommend acceptance.

**Additional Comments On Reviewer Discussion:**

One reviewer (iqRj) raised a major concern regarding the authors' motivation to study weak-to-strong deception as a phenomenon arising from competing optimization objectives. The authors' rebuttal, along with additional experiments, partially addressed this concern, leading the reviewer to raise their score after the rebuttal.

---

### Decision · Program_Chairs · 2025-01-22

Accept (Poster)